# Pinging the brain to reveal the hidden attentional priority map using encephalography

Dock H. Duncan [1,2] ✉, Dirk van Moorselaar[1,2] & Jan Theeuwes [1,2,3]

Attention has been usefully thought of as organized in priority maps – putative maps of space where attentional priority is weighted across spatial regions in a winner-take-all competition for attentional deployment. Recent work has highlighted the influence of past experiences on the weighting of spatial priority – called selection history. Aside from being distinct from more well-studied, top-down forms of attentional enhancement, little is known about the neural substrates of history-mediated attentional priority. Using a task known to induce statistical learning of target distributions, in an EEG study we demonstrate that this otherwise invisible, latent attentional priority map can be visualized during the intertrial period using a 'pinging' technique in conjunction with multivariate pattern analyses. Our findings not only offer a method of visualizing the history-mediated attentional priority map, but also shed light on the underlying mechanisms allowing our past experiences to influence future behavior.

The immense complexity of our visual surroundings presents a fundamental challenge to our finite minds. Fortunately, our brains are able to take advantage of two general principles of the world to greatly simplify this perceptual problem: first, the world is highly repetitive and therefore predictable; and second, much of what we perceive at any moment can be safely ignored. The brain's propensity to automatically learn environmental regularities is often referred to as "Statistical Learning"[1]. Recently it has been claimed that uncovering the underlying correlational structure of perceptual input may be a key method by which the brain simplifies the perceptual problem space by sharpening percepts around robust predictions, thereby reducing computational costs[2–5]. Selective attention, on the other hand, is the process whereby relevant information is prioritized while task-irrelevant information is suppressed. Attention plays a crucial role in structuring perception as the majority of what we initially perceive is filtered out by this system, thereby greatly reducing the redundancy of perceptual experience[6–8].

Given the importance of these two cognitive mechanisms in shaping our perceptual experience, it may come as no surprise that these mechanisms are deeply interconnected. For instance; while driving, it is important to direct attention to some stimuli (e.g., crossing pedestrians) but little-to-no, attention to others (e.g. irrelevant billboards). When driving in unfamiliar roads, sorting relevant from irrelevant stimuli is an attentionally demanding task; however, when driving on familiar roads the experience is quite different. Through our past experiences, the visual system can be tuned in space and time to expect relevant stimuli to appear in certain places (e.g. familiar pedestrian crossings) as well as where distractors are likely to be (e.g. familiar billboards), thereby combining the cognitive tools of attention and statistical learning into an integrated system which sharpens perception to maximize performance.

The influence of past experience on current behavior in attention is known as selection history - a category which encompasses statistical learning processes as well as other history-based effects such as value-driven attentional capture or intertrial priming[9–12]. Together with top-down (goal-driven) and bottom-up (saliency-driven) attentional mechanisms, selection history is thought to converge in what is known as the attentional priority map – a real-time representation of the behavioral relevance and saliency of the stimulus present in the visual field[11–14]. While these maps can exist in any feature space, they are often

[1]Vrije Universiteit Amsterdam, Amsterdam, the Netherlands. [2]Institute Brain and Behavior Amsterdam (iBBA), Amsterdam, the Netherlands. [3]William James Center for Research, ISPA-Instituto Universitario, Lisbon, Portugal. ✉e-mail: d.h.duncan@vu.nl

studied in the context of spatial features, where attentional priority is coded as weights on a topographic representation of physical space, and attentional selection is ultimately awarded in a winner-take-all fashion to the region with the highest activity[15,16]. Selection history can then be thought of as a layer in this system, continuously up-regulating weights to locations which contained relevant information in the past, and down-regulating locations that frequently contain distracting information[12,17].

While the concept of a priority map that drives attention selection has been a prominent notion in many theories on attentional selection[8,13,15,18–21], neuroimaging research on priority maps has notably focused almost exclusively on top-down or bottom-up influences (for a selection, see[22–25]). Preliminary studies investigating the influence of selection history on the attentional priority map have noted that, unlike top-down attention, there is little evidence that the neural processes underlying the effects of selection history can be described in terms of sustained neural processes[26–29](see[30] for contrary evidence). Instead, it has been suggested that selection history effects exist on the network level, where processes of synaptic plasticity strengthen or weaken neural connections in an emergent manner[26,31,32] (see[16] for a discussion on possible neural mechanisms). Due to this latent characteristic, history-based influences on attentional selection may fall into the category of 'activity-silent' cognitive mechanisms; named such because these effects are invisible to common neural imaging techniques which measure the downstream traces of action potentials (i.e., ERP's or BOLD-responses). As a result, the study of selection history effects on the priority map has generally been restricted to the study of differences in evoked responses to predictable and unpredictable stimuli as a proxy for latent expectations in the brain[30,32,33], and their study in their pure form has remained out of reach.

Recently, in the field of working memory, a novel approach has been proposed to visualize activity-silent neural structures; synaptic theories of working memory propose that working memory may be partially (or fully) mediated by modifying synaptic weights such that remembered information is primed for reactivation in the near future[34,35]. In this energetically efficient model, memories can be retrieved in subsequent sweeps of network activity in the brain, where the stored memory will be primed to reactivate[36,37]. Critically, it has recently been demonstrated that these latent memories can also be incidentally mis-activated simply by flooding the perceptual system with sudden input; often in the form of high-contrast visual 'pings'[38,39](see[40] for an example using TMS). This salient input causes neurons to fire in discernable patterns associated with currently retained memory items, possibly due to memory-related primed neurons incidentally firing at a higher rate than other neurons (see discussion), thereby leading to systematic neural activity patterns which can be used to decode contents of otherwise activity-silent memory[41,42].

While statistically learned attentional priority and synaptic working memory are clearly distinct processes with unique underlying cognitive mechanisms, they arguably share the feature of being latent neural structures mediated by features of network connectivity at the synaptic level (see discussion for further debate). As there is no a priori reason to believe that memory-relevant primed neural structures should be vulnerable to an impulse perturbation ("pinging") while similar latent structures related to the spatial priority map should not be, we postulate that the 'ping' technique may be appropriated to visualize learned attentional priority in a behaviorally independent manner. Such a finding would both inform the neural mechanisms underlying history influences on attentional selection as well as represent the first time the latent attentional priority map was imaged in a neutral way using a task-irrelevant ping.

In the current study, we employed the additional singleton task[43] with imbalanced target distributions, a paradigm that can be used to implicitly train participants to expect relevant information (i.e., targets) to appear in certain regions of space[14,44–46]. We then showed that while the ongoing EEG signal did not contain information regarding the current high-probability location, we could robustly decode this high-probability target location based on the ping's evoked response. Control analyzes showed that this decoding could not be attributed to temporal confounds or eye movements, suggesting instead that the ping-evoked neural responses revealed a latent, implicitly learned spatial bias operating as a silent layer of the attentional priority map.

## Results

### Behavior: Flexible learned prioritization at high-probability target locations in a changing environment

Participants ($N = 24$) performed a variant of the additional singleton task[43](Fig. 1a), with sequences of biased blocks where the target appeared with a higher probability at specific locations (i.e., four out of eight possible target locations served as high probability target location throughout the experiment; Fig. 1a) intermixed with neutral blocks without a spatial target location imbalance serving to reset the learned priority landscape (Fig. 1b; see methods for full details). Critically, on half of all trials, participants would encounter task-irrelevant high-contrast visual 'pings' in the intertrial period in between visual searches (see Fig. 1c, d).

As visualized in Fig. 2a–c, behavioral results indicated that participants were sensitive to the distributional properties of the targets across the various high-probability locations throughout the experiment - as indicated by faster responses when targets appeared in high probability target locations relative to all other locations ($t(23) = 10.62$, $p < 0.001$, $d_Z = 2.17$, 95% CI [40.72, 60.48]; Fig. 2a). Furthermore, a reliable intertrial target location effect was observed when the target location repeated from one trial to the next ($t(23) = 5.956$, $p < 0.001$, $d_Z = 1.22$, 95% CI [37.21, 76.82]; Fig. 2b). Controlling for this intertrial priming effect, the observed speed up of RT's at high probability locations remained highly reliable after excluding all trials where the target repeated from one trial to the next ($t(23) = 9.882$, $p < 0.001$, $d_Z = 2.02$, 95% CI [29.56, 47.25]; 18% of all trials excluded, Fig. 2a, c exclude target repetitions). Additionally, in line with selective changes in attentional priority as a function of the introduced statistical regularities, trials in which a distractor was presented in the high probability target location had especially slow response times, indicating that distractors interference was more pronounced when the distractor appeared in location participants had been trained to expect targets to be present ($t(23) = 3.442$, $p = 0.002$, $d_Z = 0.7$, 95% CI [10.41, 41.79]; Fig. 2c).

To examine whether the learning effect differed across experimental phases, a repeated measure analysis of variance (ANOVA) within subjects' factors target location (high probability vs. low probability) and experiment phase (high probability location 1–4, see Fig. 1b) was conducted. This yielded two main effects (all $F$'s > 8.7, all $p$'s < 0.001, all $\eta^2 > 0.033$), but no interaction ($F(3, 69) < 1$; $BF_{10} = 0.073$), indicating that the observed speed up at high probability locations was invariable throughout the experiment (all $t$'s > 2.2, all $p$'s < 0.036, all $d_Z > 0.46$; see Supplementary Fig. 1). To further test whether the neutral blocks led to extinction of previously acquired attentional bias, mean reaction times were compared in the neutral blocks between trials in which the target appeared at the high-probability location of the previous block with all seven other locations. No statistically significant difference was found between these trial types, indicating a successful extinction of bias ($t(23) = 1.444$ $p = 0.162$; displayed per-block in Supplementary Fig. 1).

Out of the 24 participants, 11 indicated that they noticed targets were presented more frequently at one location than any other. Seven out of these 11 participants were also able to correctly identify the HP location of the preceding block. When excluding these 11 participants, our remaining dataset continued to show strong target enhancement

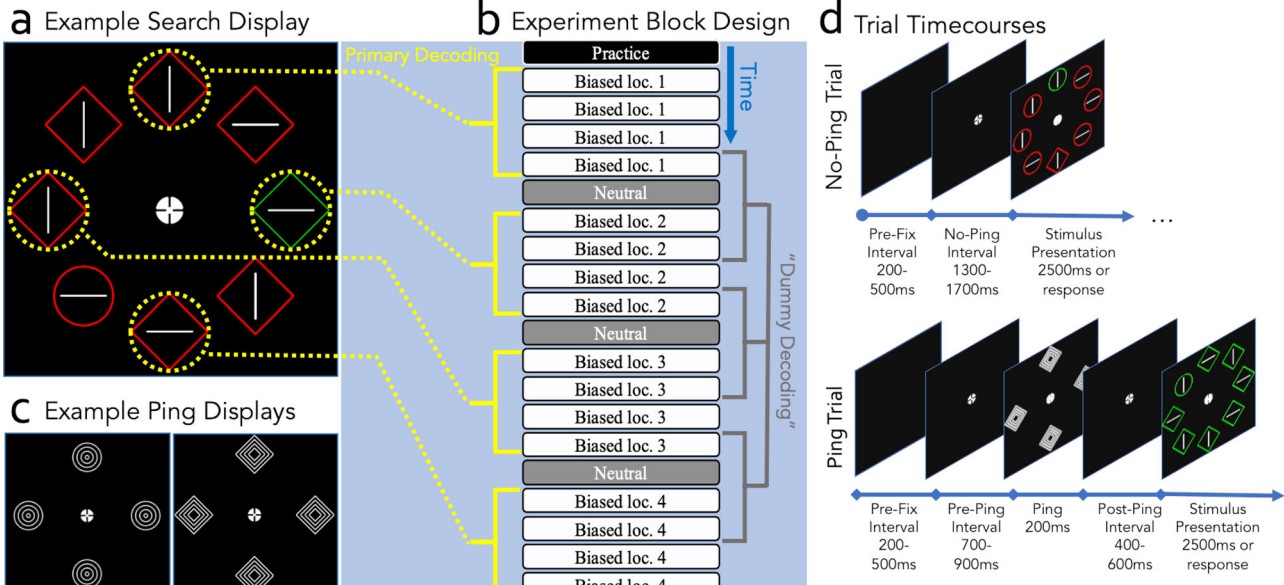

**Fig. 1 | Trial timecourse and experiment design. a** Example of a stimulus display. Note that the yellow markings were not present in the actual experiment. Participants were tasked to find a uniquely shaped shape singleton and report the embedded line's orientation using the 'z' key for horizontal or '/' key for vertical. Colors and shapes randomly varied between trials, keeping the target characteristics unknown. Salient color singletons served as distractors in 2/3 of the trials. **b** Experiment block order and decoding regime. Biased blocks had one significantly more likely target location (37.5% of trials), while neutral blocks had no high-probability location, distributing targets equally among eight locations. Participants encountered four biased blocks followed by a neutral block. The subsequent four biased blocks used different high-probability locations, cycling among four (top, bottom, left, and right) with the order counterbalanced across participants

(the order shown in the figure is just one example). The rightmost panel presents the 'dummy decoding' regime: as a control analysis, decoders received fake category labels representing random time intervals without a consistent high-probability location. Temporal features of EEG signals were preserved while removing categorical features of a shared high-probability location. **c** Examples of the two types of pings used in the experiment. **d** Timecourses for no-ping (top) and ping (bottom) trials. Pre-stimulus periods were time-matched (1500–2200 ms). Trials began with a black screen, followed by a fixation dot. In half the trials, a salient ping appeared for 200 ms. The stimulus display required participants to identify a unique shape and report the line orientation, lasting until a response or 2500 ms. Trials without pings still recorded a trigger event for baselining in the no-ping decoding analysis.

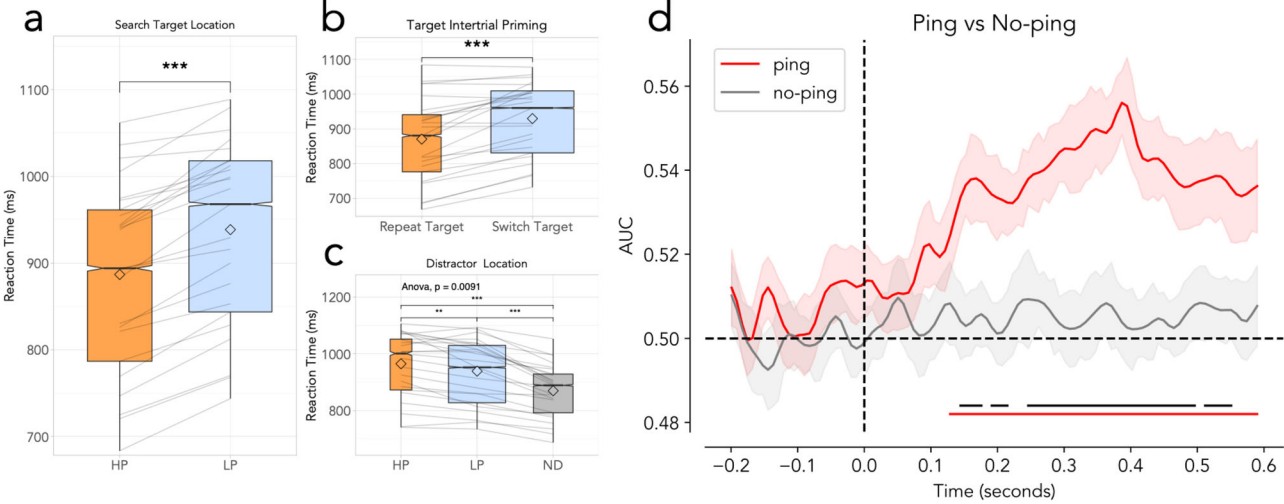

**Fig. 2 | Behavioral and decoding results (*N* = 24).** In box plots: grey lines indicate individual participant results; shaded box extends over IQR; the middle line represents mean; whiskers extend to mini/maximum values; diamonds indicate condition means; notches indicate confidence interval of median; plot size indicates data volume; all stars represent results of two-sided, preregistered t-tests; * = $p < 0.05$; ** = $p < 0.01$; *** = $p < 0.001$. **a** Participants were faster to respond to targets at high-probability (HP) locations than at low-probability (LP) locations.
**b** Participants were faster when targets were presented at the same location on sequential trials (repeat trials) than when they switched to a new location (switch target). **c** Participants were slower when distractors were present than when they

were absent. Participants were, additionally, especially slow when distractors were present at the HP target location. **d** Decoder results comparing ping and no-ping trials. Data was baselined in the −200 to 0 ms window pre-ping onset. Shaded areas represent the standard error of participant means. Lines are smoothed using the scipy function gaussian_filter with an alpha of 1.5. Lower red bars represent significant clusters identified where decoding of ping trials was above chance. Black bars represent clusters identified in which ping and no-ping decoding differed significantly. Posthoc analysis additionally showed that all four locations contributed to this above chance decoding (see Supplementary Fig. 5 for the confusion matrix, as well as decoding without boosting as per preregistration).

at the high probability location ($t$ (12) = 6.751, $p$ < 0.001, $d_Z$ = 1.9, 95% CI [25.27, 52.76]) suggesting that our observed results were not solely driven by a subset of aware participants (see Supplementary Fig. 2 for the decoding results from the next section excluding these same 11). While this does not rule out the possibility that participants had some explicit knowledge regarding the underlying manipulation[47,48], it does appear that both behavioral and decoding results were consistent regardless of reported awareness levels. These results further support recent findings that participants utilize statistically learned attentional strategies even regardless of explicit awareness[49].

### Decoding: Visual pings reveal the priority landscape in anticipation of search display onset

After having validated that locations with a higher target probability were prioritized over other locations via reaction time measurements, we set out next to examine whether this prioritization could be revealed in anticipation of search display onset, and, critically, whether such decoding was dependent on the presentation of high-contrast visual 'pings'. For this purpose, we trained a classifier on the response pattern of 64 electrodes using all four high-probability target location as classes (see Methods for full detail) separately for ping and no-ping trials. As visualized in Fig. 2d, we observed robust above chance decoding of the high-probability target location, but critically only from the evoked activity elicited by visual pings. By contrast, in the absence of a visual impulse, ongoing EEG signals appeared to contain little to no information about the current high-probability target location. Indeed, cluster-based permutation tests across time confirmed that decoding in ping trials not only reliably differed from chance, but also from no-ping trials. The observed high probability target location decoding following visual pings is in line with the idea that learned attentional priority is encoded in a latent layer of the attentional priority map, mediated by dynamic changes of synaptic weights underlying spatial attention networks leading to priming of learned responses in preparation for new sensory input. The current results show that it is then possible to decode the weights of this otherwise activity-silent spatial priority map via a salient, task-irrelevant ping. However, before reaching this conclusion, a mix of several pre-planned and unplanned alternative explanations were investigated (see also Supplementary Fig. 6 for an additional pre-registered ERP analysis. While alpha lateralization was not a primary concern for this study as these effects are only robustly seen for horizontally presented stimuli - while the current study also included laterally presented stimuli - for a further analysis of pre-stimulus alpha focusing on horizontal high-probability conditions using this dataset, see[50]).

### Decoding effects cannot be explained by temporal correlation

A concern in the present design is the blocked nature of the high probability distractor location, such that classification labels not only signaled a unique spatial high probability location but also temporally separate phases in the experiment (see Fig. 1b). While the absence of reliable decoding in the no-ping trials rules out that decoding is driven by oscillatory temporal artifacts, it is nevertheless possible that the evoked response elicited by the ping varied over time resulting in spurious decoding[51–53]. To explore whether the ping-evoked response produced meaningful decoding, in an unplanned control analysis we repeated the main analysis, but rather than classifying high-probability target locations, the model was trained to decode the position of the target in the preceding search display. This analysis included all eight possible target locations and additionally included the neutral blocks. Although less pronounced, this analysis again resulted in selective decoding following ping onset, and critically this decoding could not be explained by temporal artifacts as trials were now randomly sampled from various time points (Fig. 3a). To further rule out temporal structure as a confound, in another unplanned control analysis, we

explored whether it was possible to decode distinct temporal phases in the experiment. That is, we artificially split the experiment in three separate phases, with each phase containing the final block in a high probability location sequence, the subsequent neutral block, and the first two blocks of the following regularity ('dummy decoding' in Fig. 1b). We chose these blocks specifically to minimize overlap with the high-probability conditions while matching trial volume as closely as possible to the original decoding (224 trials). Additionally, we chose to make this window overlap slightly more with the subsequent high-probability condition than the preceding one as there was a chance that participants would be in the act of un-learning the previous high-probability for some time after the regularity was no longer present[54–57]. We sought to proactively counteract such lingering biases by sliding the dummy window more in favor of the second high-probability condition. When these 'dummy' labels were passed to the exact same decoding pipeline as in the preceding analyzes decoding collapsed, never deviating significantly from chance level for both ping and no-ping trials (Supplementary Fig. 3A). Together, these analyzes suggest that the observed decoding was indeed driven by learned latent attentional biases in response to the high probability location manipulation, and the following analyzes treat the results as such.

### Decoding was driven both by statistical learning and inter-trial effects

It is well known that in visual search response times are affected by intertrial priming in a similar way as statistical learning affects response times[58,59]. Indeed, consistent with previous studies on inter-trial priming, the current experiment also shows that response times were significantly faster when the target location repeated from one trial to the next (Fig. 2b). Critically, the ping which was presented during the intertrial interval of the current trial was able to also retrieve which location contained the target location on the preceding trial (Fig. 3a). Therefore, due to our experimental design in which there is a higher proportion of ping trials preceded by targets at the high probability location, it is possible that the observed decoding does not reflect learning across longer time scales, but instead solely reflects intertrial priming effects. To determine whether this was the case, in an unplanned analysis we repeated the main analysis after excluding all trials in which the preceding trial contained a target at the current high-probability (HP) location (37.5% of trials, called HP-exclusionary). Under these conditions, ping trials continued to show significant decoding of the high-probability location (Fig. 3b). A further exploratory analysis was done to test whether decoding would remain high when decoders were trained *only* on pings following trials in which the target was at the current high-probability location (called HP-only decoding). Decoding continued to remain high despite a significantly lower trial count than the HP-exclusion analysis, indicating that the decoding contribution was approximately equal between HP trailing and non-trailing trials despite the trial imbalance (Fig. 3b). Importantly, while intertrial effects can be separated from statistical learning effect via trial exclusions, statistical learning effects cannot be separated from intertrial effects as they are present in all trials. Decoding in the HP-exclusionary condition of Fig. 3b should thus be thought of as a combination of statistically learned enhancement and serial biases, while Fig. 3a better reflects the pure intertrial effect. Overall, these results match the behavioral results, where significant effects of intertrial priming were found on top of statistically learned spatial enhancement, and indicate that the ping is able to retrieve the priority landscape both induced by serial priming from the previous trial and statistical learning across longer timescales.

### Decoding effects cannot be explained by systematic shifts in gaze position

A common concern in multi-variate EEG analyzes is the influence of eye movements on decoding[60]. For example, the position of the eye can

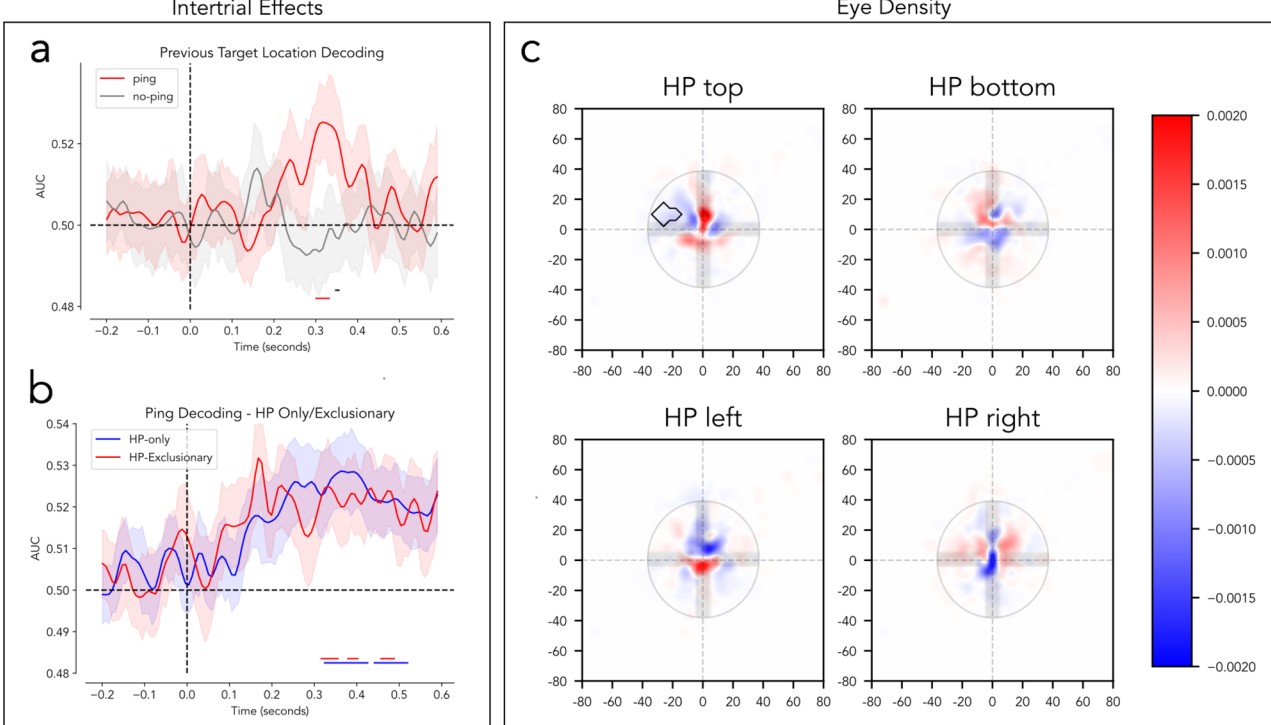

**Fig. 3 | Control Analyses.** Solid colored bars below decoding results indicate significant clusters as identified via permutation tests (see Methods). All results were smoothed using a gaussian filter. Shaded areas represent standard error of mean. **a** Decoder results when passed preceding trial target location as labels instead of current high-probability target location. Note that this decoding took all 8 locations as factors and included the neutral blocks. This analysis also included trials that could have been at the current high-probability location (32% of total trials). Trial averaging for this analysis was done over five trials (see Decoding methods). **b** Decoding results when decoders are trained and tested only on ping epochs which followed a trial in which the target was at a HP location (blue) or excluding all such HP trailing epochs (red). Note that due to low trial counts, no trial averaging was possible for HP-only decoding (See Methods). **c** Eye-density plots showing average eye position in the 600 ms window following pings. Red represents regions with higher densities than combined average of other three conditions. Blue represents regions with lower densities. Black outlines indicate significant clusters identified using permutation tests (see methods). X and Y-axis ticks indicate pixel distance. Color bar indicates proportion out of 1. The fixation dot has been superimposed over the densities for reference. See Supplementary Fig. 4A for an un-baselined towardness analysis of eye data in the intertrial period.

influence neural decoding when participants make (small) eye movements, affecting how the image is projected to early visual cortex and thus possibly leading to distinct patterns of activity[60,61]. There is some reason to believe that if eye movements were significantly influencing decoding, then they should have also influenced decoding in no-ping trials as the fixation cross remained present on the screen, thus leading to patterns of activity which should have been picked up by the decoders. This assertion is further supported by the fact that foveal receptive fields are much denser than parafoveal fields, meaning minute movements should have evoked an outsized response to the foveal fixation point in comparison to the peripherally presented ping stimuli[62,63]. Despite this reasoning, and to address the issue of eye movements, we systematically investigated the extent to which the decoding results could be attributed to eye-related artifacts. First, we examined whether gaze was shifted relative to the high-probability target locations. To do this, we calculated average gaze position for each of the HP trials across participants to form density maps of gaze positions per condition. As is evident from the heat maps of gaze density, this analysis yielded no evidence that gaze was systematically shifted towards the high-probability target location (Fig. 3b). The only significant cluster found was a negative cluster to the left of the fixation dot when the high-probability location was the top of the screen (cluster-based permutation test, $p < 0.05$). Excluding this condition from the main analysis did not change the overall pattern of results. To build on the results of the heatmap analysis, we next calculated a 'towardness score'[64] to quantify how eye position changed systematically across time in relation to the high-probability locations. This

score quantifies systematic gaze shifts as a numeric score, and is sensitive enough to both microsaccades as well as overall shifts in resting gaze position in eye tracking data[64]. The results of this analysis are shown in Supplementary Fig. 4A; eye deviations did not systematically differ from zero, indicating no systematic gaze bias towards (or away) from the high-probability locations. Furthermore, to assess whether individuals whose eyes drifted more had higher decoding scores, a correlation analysis was also done between each participants towardness score and their decoder performance over the 300–400 ms window post ping (where decoding was highest). This correlation analysis is shown in Supplementary Fig. 4B, and shows that resting eye position did not predict decoder performance.

While the heatmaps and towardness scores did not suggest any systematic eye movements across participants in relation to the high-probability location, there is still the possibility that eye movements differed between high-probability target locations, but in a way that does not generalize across participants. To test this with an unplanned analysis, we adopted the same procedure as in the preceding EEG decoding analysis, but instead entered the horizontal and vertical gaze position as features. As visualized in Supplementary Fig. 4C there was reliable above chance decoding, independent of ping presence. When decoder results from the eye-tracker-trained decoders was compared to the EEG-trained decoders, no significant correlation was found, indicating high decoder accuracy in the eye-tracker decoders did not predict high decoder accuracy in the post ping window in the EEG-trained decoders (Supplementary Fig. 4D). Because eye-movements seemed to differ systematically on a participant level, but not at the

group level, it was plausible that these drifts represented temporal artifacts rather than meaningful evoked movements. To examine this possibility, the same 'dummy' control analysis was done for the eye tracking decoding as for the EEG decoding to test for temporal confounds. Eye tracker data was passed to a decoder trained on labels with no meaningful overlap with experimental conditions (Supplementary Fig. 3B) and produced the same pattern of results as when passed meaningful labels, a result that was not found for the EEG analysis. This indicates that the decoding observed in the EEG analysis was not the result of ping-related eye confounds, but rather that above chance eye tracker decoding likely represents temporal noise in the eye tracking data.

## General discussion

How does our past experience influence future behavior, and how can we study this latent bias? We sought to answer this question on the level of attentional deployment by testing whether the ping technique (a method used previously to reveal latent working memory content) could be used to visualize the layer of the attentional priority map maintained by selection history. Consistent with an activity-silent model of learned attentional bias, here we demonstrate that when observers learned to prioritize a given location in space (as indexed by faster target selection and increased distractor interference at that location), the ongoing EEG signal did not contain information about the current high probability location. Critically, however, this otherwise hidden latent priority landscape could be revealed by inserting neutral visual ping displays in between search displays. This above chance decoding could not be attributed to temporal artifacts, nor could it be explained by systematic biases in gaze. Instead, high probability location decoding indicates that the ping technique can be used to visualize not only latent memory representations but also our latent attentional biases.

The current evidence suggests that updating local priorities across visual searches accrues extremely fast[57], which makes spatial probability learning very flexible[12]. Indeed, in the current study where the high probability location was not static but systematically varied across time, learned priority was quickly adjusted in response to a location change. One cannot exclusively attribute this observed benefit to statistical learning, however, as such statistical learning is naturally conflated with intertrial priming effects[56,65]. Interestingly, it was recently observed in monkeys and humans that intertrial effects are mediated by activity-silent mechanisms similar to those proposed for working memory[27]. Consistent with this notion, we found that the target location on the preceding trial could also be decoded in response to the onset of a visual ping. Critically, our control analyzes demonstrated that the observed high probability decoding could not exclusively be attributed to such intertrial priming effects, as decoding showed more or less identical time courses when high probability location decoding was limited to trials with a target on that location or after excluding those same trials. We thus conclude that the visual ping is able to both envision selection history effects on a very short time scale (i.e., intertrial priming) and effects that arise across longer time scales (i.e., statistical learning). These results support the notion that intertrial effects are mediated by synaptic mechanisms, as well as provide for the first time, neural evidence that statistical learning may also be mediated by such network-level mechanisms in the brain. Furthermore, these findings suggest that 'pinging' may become a useful tool for the study of both statistical learning as well as intertrial effects, and may be expanded to the study of other selection history effects (e.g. reward-based history effects[66]).

Dynamic shifts in synaptic weights have previously been proposed as a viable mechanism for spatial attention[16,18], and pinging techniques have been proposed as candidates for visualizing these network-level attentional changes in the past[40]. However, the enthusiasm for the study of these network-level influences has been blunted

by the fact that top-down attention does, in fact, produce ongoing, measurable neural activity[67–69]. Selection history effects, however, have recently been shown *not* to be driven by similar active neural mechanisms[26–29]. The current results build upon these findings by demonstrating that the ping technique can be used to reveal activity-silent history-modulated attentional bias. While these findings seem to suggest that selection history influences exert themselves at the level of synaptic weight changes, some caution is called for in this interpretation as several open questions remain over the neural mechanism's underlying ping-revealed structures as well as why these structures are usefully revealed via these transient pings.

As decoding is by itself uninformative about the underlying neural representation[70,71], the nature of these activity-silent states that the ping succeeds in visualizing remains an open question. In the field of working memory, the debate over why pings reveal otherwise hidden memory content has focused on the question of whether the decoded memories are encoded in truly latent networks, mediated by neuroplasticity, and which are then reactivated by the ping[39,72–74], or alternatively whether these memories are simply mediated by ongoing neural activity below a certain detection threshold in which case, the ping would simply serve to reduce signal variance such that these states can be visualized[75–77]. While these questions are ultimately better answered via neurophysiology, in the current case of ping-evoked decoding of learned attentional preparation the existing literature favors the former interpretation, as selection history, whether in a form of statistical learning or intertrial priming, has generally exhibited none of the characteristics of active neural processes[26–28,32]. In fact (and in sharp contrast with the field of working memory) there exists no major theory of the underlying mechanisms of statistical learning that is explicitly built on a concept of sustained, continuous neural activity. This is because, while active traces may in principle be a plausible neural mechanism for driving intertrial priming over relatively short timescales, such active mechanisms seem unlikely to drive statistically learned spatial biases which are known to persist over very long time periods ranging from minutes[54,55,57], to weeks[56,78,79]. As a result, models of statistical learning have generally resembled models of long-term memory (where synaptic models are favored for their durability over time) rather than working memory (where active firing models are more normal)[14,80–83]. Despite this, it is important to realize that the current results should not be taken as an outright refutation of any model of history-modulated attentional bias which might propose ongoing neural activity as the central neural mechanism underlying the effect. While our results are consistent with a model where there is no meaningful ongoing activity in the intertrial window, due to our use of baselining and filtering, we cannot definitively rule out that there is some activity still present. While further work is needed to clarify this point, what remains clear is that the pinging approach offers a novel tool for studying the underlying changes occurring in the brain which allow for history-mediated behavior to arise.

A further important unanswered question remains: why exactly do these otherwise hidden structures produce discernable activity patterns in response to a neutral, salient ping? In the case of working memory, it has been proposed that the observed reactivation mirrors the way that active sonar uses reflections of audio pulses to reveal underwater structures[38,39,72,73]. While a useful tool for conceptualizing the pinging technique, the reflections of audio pulses underwater is clearly mechanically very different than the interaction of visual pulses with hidden states in the brain[72]. Alternatively, it has been proposed that a latent history-dependent filter may drive decoding in working memory versions of ping experiments[37,39], an explanation which is loosely valid in our attention variant of the paradigm and which has support in various template theories of attention[18,21,84]. However, a filter match model does not necessarily imply irrelevant stimuli should also activate these templates; thus, further work must be done to justify this filter-correlated activity. We propose that the most

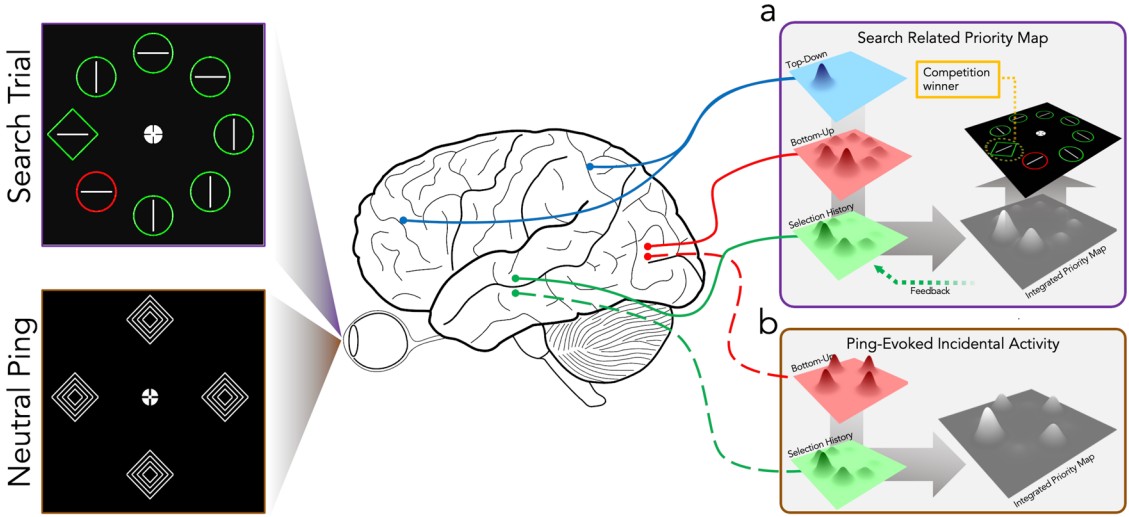

**Fig. 4 | Possible mechanisms underlying search priority and ping evoked activity.** Cells tuned to HP locations are proposed to adopt a more liberal activity threshold, thereby leading to faster correct RT's as well as distractor capture when presented in that location. Importantly, this shifted bias also leads to more incidental activity when pings are presented. This differential in incidental activity is proposed as driving ping decoding. **a** Integrated activity from top-down, bottom-up and selection history effects driving attentional selection in an example search task. Attention is directed to the region with the highest activity, in this case, the far-left location (note: the brain schematic is based on the figure used in Theeuwes et al., 2022). **b** Ping-evoked incidental activity integrating bottom-up and selection history effects. Top-down effects are absent as no task goals were present in relation to pings.

parsimonious explanation for this decoding is that the sudden, salient stimulation of the ping drives systematic mis-activation patterns by dynamically task-primed neurons (Fig. 4). Firstly, the influence of selection history is proposed to lower activation thresholds of predicted stimulus tuned neurons, this liberalization would then account both for speeded responses when targets were correctly predicted, but also increased distractor interference when present at expected target locations. Secondly, this lowered activation threshold should also lead to increased incidental firing of primed neurons when presented with the task-irrelevant ping (Fig. 4b). Incidental firing of neurons to the presentation of non-tuned stimuli is a familiar phenomenon in research using single-unit recordings (see, for instance[85],). If latent learned attentional priority is mediated by neural structures entering a 'primed' neural state via processes of neural plasticity, then the passing of irrelevant but high-contrast pings through the visual cortex may incidentally activate these primed neurons at a high rate. This rate of activation, then, would be the weak signal to which the decoders are sensitive (Fig. 4b).

Our findings also inform the debate whether learned expectations exert their influence already in advance[86] or, alternatively, only become apparent in response to sensory stimulation[3,87]. Based on behavioral studies that randomly intermix search displays with probe displays, where the latter allows one to take a peek at selection priorities immediately prior to search display onset, it has been argued that learned attentional biases are already evident pre-stimulus[46,88]. Consistent with such proactive enhancement in the spatial priority map, here the ping evoked decoding indicates preparatory spatial tuning towards the high probability location before the onset of the actual search display. At the same time, however, this proactive tuning was not evident in ongoing oscillatory activity and only became apparent in response to sensory stimulation by the visual pings. Therefore, it appears that while statistical learning proactively adjusts the spatial priority map, this priority landscape only becomes apparent after the integration of bottom-up sensory input, such as a probe display, or in this case visual pings (but see[30]).

A number of influential new theories of attention have highlighted the need to integrate selection history into models of dynamic attentional priority[10,12,89–91]. The current results advance this project by providing a novel method of visualizing the history-mediated layer of the attentional priority map while also suggesting the neural mechanisms underlying such latent biases. While priority maps have typically been studied in their influence on spatial attention, it has also been noted that they can easily be thought of as a general cognitive tool useful in the conceptualization of any goal-directed behavior[16]. Under this view, the current findings may represent just the first step in expanding the application of the ping technique to reveal latent biases in the brain. Fields of study that have previously been excluded from neuroimaging research should now reconsider the possibility of undertaking the study of latent neural states using similar pinging approaches, as much future research will undoubtedly reveal the extent of the ability of the pinging technique to reveal previously hidden structured in the brain.

## Methods

The experimental design and all analysis methods were preregistered on Open Science Framework on September 27, 2021. Preregistration can be found at https://osf.io/5vw7t. All deviations from preregistration are noted.

### Participants

This study was conducted at Vrije Universiteit Amsterdam and conformed to the Declaration of Helsinki and was approved by the Ethical Review Committee of the Faculty of Behavioral and Movement Sciences, Vrije Universiteit Amsterdam. All participants provided written informed consent prior to participation and were compensated with 25 euros or course credits. All participants indicated normal or corrected-to-normal vision and reported no history of cognitive impairments. Based on previous pinging studies on working memory[38,39,92] we included 24 participants (17 female, mean age 24) in our final dataset after replacement of eight participants. No statistical method was used to predetermine the sample size. Two participants were replaced because of accuracies 2.5 standard deviations (SD's) below the group average on the behavioral task; two as a result of overall response times slower than 2.5 SD's from the group mean; two for producing low-quality encephalography as revealed through visual inspection, and two for failing to maintain fixation during the task as revealed through the electro-oculogram and eye-tracking analysis (saccades detected on more than 30% of trials, see eye-tracing

acquisition and preprocessing below). Sex was not taken as a factor for analysis as we had no a priori reason to expect sex differences in cognition.

## Apparatus and task

Participants were seated in a dimly lit sound-attenuated room with a chinrest 60 cm away from a 23.8 inch, 1920x1080 pixel ASUS ROG STRIX XG248 LED monitor with a 240 hz refresh rate upon which all stimuli were presented. The behavioral task was programmed using OpenSesame[93] and utilized functions from the Psychopy library of psychophysical tools[94]. EEG data were recorded with default Biosemi settings at a sampling rate of 512 Hz using a 64-electrode cap with electrodes placed according to the international 10–10 system (Biosemi ActiveTwo system; biosemi.com) with two earlobe electrodes used as offline reference. Vertical (VEOG) and horizontal EOG (HEOG) were recorded via external electrodes placed ~2 cm above and below the right eye, and ~1 cm lateral to the left and right lateral canthus. Eye-tracking data were collected using an Eyelink 1000 (SR research v. 4.594) eye tracker tracking both eyes. Participants used a headrest for stability positioned 60 cm from the screen. Sampling differed between participants between 500, 1000, and 2000 hz due to the different EEG labs used having different versions of the Eyelink 1000 with variable sampling limits. All participant data was later standardized to 500 hz during preprocessing. Calibration was done before the first block and at the halfway point for all participants. For some participants, additional calibrations occurred due to subtle changes in resting position in the chinrest or other factors which caused noticeable drift in the calibrated signal.

The task was a modified version of the additional singleton task with underlying stimulus regularities known to provoke statistical learning[43,95]. All stimuli were presented on a black background. Each trial started with a blank screen randomly jittered between 200–500 ms in length. A circular fixation point (40 px diameter, ~1.1°) was then presented alone between 1300–1700 ms at the center of the screen. The fixation point design was taken from Thaler et al. [96]. who showed their fixation dot design provided the most stable fixation results from a range of possible designs. Participants were informed to maintain fixation throughout the trial and not to saccade towards any peripheral stimuli. Participants received feedback when the eye tracker detected a fixation deviation above 1.5° away from fixation in the form of a low-volume audio beep.

During the fixation period, on 50% of trials a high-salience visual 'ping', which appeared at a time point randomly jittered between 700–900 ms after fixation onset, was presented for 200 ms (Fig. 1c). Pings were comprised of four white (rgb 0,0,0) high contrast shapes, either all diamonds (diagonal length 116 pixels; ~3°) or all circles (diameter 90 pixels; ~2.4°), presented in the four cardinal directions (up, down, left and right) on the same locations where subsequent search shapes appeared (see below). The ping shapes comprised of a large outer shape with three smaller shapes recursively embedded within the outer shape. We chose to use pings that closely resembled the search targets in location and form as this is typically done when this procedure is applied to investigate working memory[38,39,92,97] (but see[40,98] for exceptions). Following ping offset, the fixation point remained on screen for a randomly jittered duration of 400–600 ms. To contrast ping and no-ping trials, a ping trigger was created on both trial types, but on no-ping trials, these triggers did not correspond with an actual ping presentation; instead, the screen remained blank except for a fixation dot until the next trial began.

An additional singleton search display[43] was next presented, which remained on screen for 2000 ms or until participants provided a response (Fig. 1a). Each search display contained eight equidistantly placed shapes on an imaginary circle (radius 185 pixels; ~4.8°) centered on fixation. The shapes, which could either be circles (diameter 90 px, ~2.4°) or diamonds (82 x 82 px or ~ 2.1° x 2.1° square rotated 45

degrees) and could either be colored red (rgb 255,0,1) or green (rgb 0,128,0), all contained a white (rgb 255,255,255) line (70px; ~1.8°) oriented horizontally or vertically bar. On each display, one shape was unique from the rest (either one circle among seven diamonds or vice versa) and participants were instructed to report the orientation of the line inside this target shape via button press on a standard keyboard ('z' for horizontal lines; '/' for vertical lines). On a subset of trials (70%) one of the homogenous shapes was assigned a unique color (e.g., if the shapes were green, the distractor was red, or vice versa) rendering it a colored singleton distractor. The sole purpose of these distractors was to make the task more challenging, thereby making the target probability manipulation less apparent (see below). Distractors' presentation was controlled such that a distractor appeared in each of the eight locations exactly the same number of times per block.

Critically, target locations were not selected at random, but instead, the experiment was structured such that for several experiment blocks in a sequence, one location would be more likely to contain the target than the other locations (see Fig. 1a). Specifically, in high probability target blocks, one location was disproportionately more likely to contain a target, with targets appearing at this high-probability location on 37.5% of trials, making them 4.2 times more likely to contain a target than any of the other, low-probability locations. To be able to decode the high-probability target location, high-probability locations did not remain static for an entire experimental session, but instead periodically shifted across the cardinal locations (up, down, left, and right). Specifically, in every four blocks the high probability location changed to a new location (order counterbalanced across participants), with every change being preceded by a neutral block, in which targets appeared in each of the eight locations with equal probability (see Fig. 1b). These neutral blocks served to, at least partly, unlearn the acquired spatial priority settings such that observers entered a new sequence of learning blocks without robust lingering attentional biases[54–57].

Each of the 19 experiment blocks and the preceding practice block consisted of 56 trials. Breaks were offered in-between experimental blocks where participants were informed of their progress, accuracy, and average reaction time and were encouraged to rest their eyes. Following the completion of the experiment, the participant was asked to answer an additional four debrief questions (for three participants the debrief was collected verbally). Firstly, they were asked if they noticed the target tended to appear in certain locations more frequently than others. Secondly, they were asked to indicate where they believed the target was most frequently present on the final experiment block which they had just completed. If they were not sure they were instructed to provide a best guess. Thirdly they were asked if they felt any of the other seven locations at some point in the experiment was more likely to hold a target. Finally, the participant was asked if they had ever performed a task similar to the one they had just finished.

## Analysis

All analyzes involving EEG and eye-tracking data were done using custom Python scripts. All behavioral analyzes were programmed using R. All preprocessing and analysis scripts are available online and can be downloaded at https://github.com/dvanmoorselaar/DvM. Functions from the open-source MNE analysis package were heavily used and central to our analyzes[99]. All analysis steps followed those listed in our preregistration unless otherwise stated.

## Behavioral preprocessing

If a participant's overall accuracy was 2.5 SD's below group mean, then that participant was excluded from our dataset and replaced. The average accuracy after exclusions was 89% (range 81–97%). Next, participant data was restricted to correct trials, and mean and SD based on each individual participant's RT data were then calculated. Trials 2.5 SD's faster and slower than each participant's average RT were

excluded, resulting in an average of 1.1% of abnormally slow/fast trials excluded per participant. The group average reaction time (RT) was then calculated and participants 2.5 SD's away from this mean were excluded and replaced. Combined exclusion of incorrect responses (~11%) and data trimming (~1%) resulted in an overall loss of approximately 12% of total trials

### EEG acquisition and preprocessing

Following re-referencing of all EEG data to the average of the two earlobe electrodes, the data were high-pass filtered at 0.01Hz using Hamming windowed FIR filter to remove slow signal drifts. Continuous EEG was then epoched from −700 to 1100 ms relative to ping display onset, or relative to matching timepoints on no ping trials, with trial rejection procedures being limited to a smaller time window (i.e., −200–600 ms). Prior to subsequent trial rejection and artefact correction, malfunctioning electrodes marked as bad by the experimenter during recording were temporarily removed. First, EMG contaminated epochs were identified with an adapted version of an automated trial-rejection procedure as implemented in Fieldtrip[100]. To specifically capture muscle activity we used a 110–140 Hz band-lass filter and allowed for variable z-score thresholds per subject based on within-subject variance of z-scores[101]. Moreover, to reduce the number of false alarms, rather than immediate removal of epochs exceeding the z-score threshold, the algorithm first identified the five electrodes that contributed most to the accumulated z-score within the time period containing the marked EMG artefact (We chose to deviate from our preregistered method here as the new interpolated method allowed us to preserve more data. Results were also analyzed using the pre-registered exclusion criteria and did not differ in the pattern of results.). Then in an iterative procedure, the worst five electrodes per marked epoch were interpolated using spherical splines[102] one by one, checking after each interpolation whether that epoch still exceeded the determined z-score threshold. Epochs were selectively dropped when after this iterative interpolation procedure the z-score threshold was still exceeded. Second, Independent Component Analysis (ICA) as implemented in the MNE (using the 'picard' method) was fitted on 1 Hz high pass filtered epoched data to remove eye-blink components from the cleaned data. Third, manually marked malfunctioning electrodes were interpolated using spherical splines.

Lastly, we excluded trials in which a saccade was detected by the eye tracker, or significant drift was recorded through the course of the trial. To detect drift in our eye tracking data, we first baselined our eye tracking data on the 300 ms pre-ping window. We then observed the maximum deviation from zero measured in segments of data 40 ms long for each trial. If this maximum exceeded 1° then the trial was marked for exclusion (~8% of data per participant). If the number of trials excluded due to saccades or drift exceeded 30% of the total trials, their eye tracking data was further examined. If it was revealed that eye tracking data was exceptionally low quality, then their eye tracking data was ignored and their data was cleaned based on the HEOG recordings instead (one participant had their data treated in this way). If their eye tracking data was found to be of high quality, then the participant was excluded from the analysis and their data replaced with a new participant (two participants were excluded in this way). For participants with no reliable eye tracking, we identified trials with sudden jumps in the recorded EOG facial electrodes using a step method with a window of 200 ms, a step size of 10 ms and a threshold of 20 μV, and excluded these trials.

### Eye-tracking acquisition and preprocessing

One participants eye data was excluded due to poor quality tracking (see above). For the remaining 23 datasets, eye data was first converted from the native EDF format to the ASC format before further processing. The eye-tracking data was then epoched around the ping onset event, or relative to matching timepoints on no ping trials, including

−200 ms before onset and 600 ms after onset to match EEG data epochs. Blinks were then identified in the eye-tracking signal and linearly interpolated using custom scripts. A 200 ms pad was applied before and after any identified blink time window to ensure movement artifacts before and after blinks were not accidentally preserved. All eye data was then baselined relative to the 200 ms window prior to ping onset when participants held fixation. If the sampling rate of the participant was above 500 hz, their data was next down-sampled to 500 hz using scipy resample function. Eye tracking data was then trial matched to the processed EEG data such that only trials included in the EEG analysis were also included in any eye tracking analysis.

### Eye-density calculation

For each participant, eye data from the 600 milliseconds post ping were first separated between the four HP conditions and concatenated. Next, each datapoint was rounded to the nearest quarter of a pixel and centered around zero. Density was then calculated for every quarter pixel by dividing the count of the number of timepoints at each (x,y) location by the total number of timepoints. Next, density differences were calculated by cycling through each of the four conditions and subtracting every cell of the current density matrix by the averaged density matrix of the other three conditions. The group average density figure was then generated by averaging across the 24 individual density matrixes for each HP condition. Finally, a cluster permutation test was done for each of the four locations, across the 24 datasets (see statistics).

### Eye towardness calculation

The eye towardness analysis shown in Supplementary Fig. 4A is based on that used in van Ede et al.[64] In this preregistered analysis, un-baselined eye tracking data was used. Data was first epoched in a -700 ms to 600 ms window centered around (hypothetical) ping onset. This window extended from the earliest fixation onset to the earliest next search trial onset, and thus represents the majority of the inter-trial window on every trial. Trials were then split into ping and no ping trials to be analyzed separately. Following the methods of van Ede et al.[64]. Experiment 2, we next separated our trials into those in which the high probability location was on one of the vertical positions (top or bottom), or one of the horizontal positions (left or right). Separately for each participant, for every individual timepoint we then averaged the x-axis channel values for the horizontal trials and the y-axis channel values for the vertical trials (see van Ede et al.[64] eye-tracking analysis for further information) to get the average x or y-axis behavior through the intertrial period. Next, we subtracted the right from the left values, or the top from the bottom values, and divided by two to get a difference score, with positive values representing the average eye positional bias towards the high probability location, and negative values representing bias away. Next, we further divided these values by the actual possible target locations and multiplied by 100 to get a percentage deviation towards the target locations. Finally, we averaged between our vertical and horizontal conditions to get a general towardness score for ping and no-ping trials.

### Decoding analysis

To test whether the ping evoked discernable activity based on the different high-probability locations used across various blocks, we applied multivariate pattern analysis (MVPA) using a cross-validated linear discriminant analysis (LDA) on ping and no-ping trials separately with all 64 electrode channels as features and block high probability locations as classes. EEG data was first baselined in a window from -200 to 0 ms prior to ping onset trigger (for no-ping trials, no stimuli was actually presented but instead the screen remained blank except for a fixation point), and subsequently down-sampled to 128 Hz. While not preregistered, to further increase the signal-to-noise ratio in our data[103], we adopted a trial-averaging approach and subsequently

transformed our data using principal component analysis (PCA). Specifically, we averaged over three trials of the same exemplars, after which the data was PCA transformed in each cross-validation run (see below) with a model fitted on the training data only. The main decoding results without these additional signal-to-noise boosting effects are included in Supplementary Fig. 5.

The data was next randomly split into 10 equally sized subsets where each class (i.e. the four possible high probability locations) were selected equally often in each condition so that training would not be biased towards any one class. Next, a leave-one-out procedure was used such that each classifier was trained on nine folds and tested on the one excluded fold until each fold was tested once; thus, ensuring that training and testing never occurred using the same data. Classifier performance was then averaged across the ten folds. We used an Area Under the Curve (AUC) approach to rate classifier performance, which is an approach considered a sensitive, nonparametric and criterion-free measure of classification performance[104]. Using AUC, a rating of 0.5 is considered chance level classification. An analysis score was collected for every time point, showing how decoding performance fluctuated across time. We chose to decode over the time period from -200 ms before ping onset until 600 ms after ping onset (or 400 ms after ping offset, the minimum amount of time that could pass before a search display appeared, see Fig. 1d).

### ERP analysis
An additional event-related potential (ERP) analysis was conducted to investigate whether the ping displays produced a lateralized evoked component, such as the N2pc, which is known to indicate lateral attentional capture[105–107]. While the results did not produce anything of real interest, this analysis was included in the preregistration and thus is included as Supplementary Fig. 6 for completeness.

Two separate ERP analyzes were conducted for ping and search displays. Data was 30 Hz low-pass filtered and baseline corrected using a 200 ms window prior to ping or search onset We first investigated whether an N2pc component could be found during search as a sanity check. Trials were thus restricted in our search analysis to those in which the target appeared on the left or right side of the horizontal midline. Furthermore, interference from the distractors was controlled by restricting trials of interest to those in which the distractor was present on the vertical meridian or trials in which no distractor was presented at all. For our analysis of the ping data, the primary factor of interest was block HP conditions, so analysis was restricted to trials in which the HP location was either the left or right locations in space, as there was no reason to expect top and bottom conditions to produce a perpendicular lateralized component. For each analysis, waveforms were computed for ipsilateral and contralateral scalp regions relative to the relevant horizontal factor using the PO7/PO8 electrodes of interest. Data was then 30 Hz low-pass filtered and baseline corrected using a 200 ms window prior to ping or search onset.

### Statistics
RT and accuracy were analyzed using simple within subject, two-sided t-tests and analyzes of variance (ANOVA's). For all frequentist statistics, the appropriate tests for assumptions were carried out (e.g. Shapiro-Wilke, Mauchly etc.). In the case that a test failed an assumption check, this violation would have been reported and the appropriate revised test carried out. In case of insignificant results, where applicable, we run Bayesian equivalents of the above specified tests using the default prior settings of the JASP analysis toolbox[108]. Decoding scores were evaluated for both ping and no-ping trials via cluster-based permutation tests and paired sampled two-sided t-tests with cluster correction ($p = 0.05$ and 1024 iterations) using MNE functionality[99]. A similar permutation test was used to compare whether the difference between decoding on ping and no-ping trials was significant across time.

### Reporting summary
Further information on research design is available in the Nature Portfolio Reporting Summary linked to this article.

## Data availability
The anonymized raw EEG and behavioral data generated in this study have been made publicly available on OSF (https://doi.org/10.17605/OSF.IO/V7YHC; https://osf.io/v7yhc/).

## Code availability
All scripts related to EEG preprocessing and analysis, behavioral data preprocessing and analysis, as well as the experiment code have been made available on the project's OSF page (https://osf.io/v7yhc/) and the associated toolbox's Github (https://github.com/dvanmoorselaar/DvM).

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

## Acknowledgements
This research was supported by a European Research Council (ERC) advanced grant 833029 – [LEARNATTEND] awarded to J.T. The authors wish to thank Freek van Ede for his invaluable guidance in our density and towardness eye tracking analyses and Johannes Fahrenfort for his careful consideration of our decoding results and helpful advice. The authors would also like to thank Clayton Hickey for his expertize in identifying our need for a temporal correlation control analysis, and his time in helping us think up a control analysis.

## Author contributions
Original idea by D.H.D., D.v.M., and J.T. Experiment programmed by D.H.D. and D.v.M. Data collected by D.H.D. Analysis programmed by D.H.D. and D.v.M. Manuscript drafted and revised by D.H.D., D.v.M. and J.T.

## Competing interests
The authors declare no competing interests.
