## [Peer Review File · Nature Communications]

Reviewers' Comments:

Reviewer #1:

Remarks to the Author:

When we frequently perform a visual task with statistical regularities (e.g., looking for our keys on the way out the door; the keys are usually on the kitchen table, but sometimes your toddler moves them to the chair), behavior improves in its speed and accuracy based on the use of these regularities. In this manuscript, the authors test the possibility that spatial regularities learned during a visual search task are instantiated in a 'silent' adjustment of synaptic weights that support neural priority maps of visual space. They adopt the 'pinging' paradigm frequently used in visual WM experiments (e.g., Wolff et al, 2015) to measure the impact of a putative latent state of the neural priority map on visual processing of task-irrelevant visual stimuli. They find that, when a location is statistically more likely to contain a target stimulus, this location can be decoded based on evoked EEG activity patterns in response to a task-irrelevant ping stimulus. The authors conduct some important control analyses to rule out contributions from, e.g., eye movements. Overall, the authors conclude that learned probabilities of target locations on a visual search task are neurally instantiated via activity-silent mechanisms, consistent with a rapid reshaping of synaptic weights.

I really like this paper – the idea is simple but profound, and is one of those that made me think "how did I not think of that??" The results are intriguing, and will certainly be of substantial interest to the readership of Nature Communications. I have several comments about the analyses presented by the authors (and the preregistered analyses), and other aspects of the manuscript. I expect these can be addressed with the data in-hand.

Major:

1. I'm having a bit of trouble understanding the "dummy decoding" analysis illustrated in Fig. 1, and reported in Supp. Fig. 3A. The goal of this analysis is described as ruling out the possibility that subtle changes in ERPs over the course of the experiment session could explain the ping-evoked decoding observed independent of learned spatial regularities. The authors thus sample contiguous blocks of trials that span 2 blocks of one biased location, a neutral location block, and 1 block of another biased location. The authors describe in the caption of Fig. 1B that the authors use "fake category labels" – I'm interpreting this to mean that they arbitrarily assign each of the 3 dummy sets of blocks a number (1-3) and attempt to decode this value. The authors motivate the use of 4 adjoining blocks based on the importance of matching trial numbers, but the current scheme seems like it would still result in a biased sample of trials (more trials with 1 biased target location than any other). Could the authors instead sample 1.5 blocks before/after the neutral blocks? Perhaps the lack of significant decoding observed in Supp Fig. 3A suggests that this concern is moot, but I thought I'd raise it, as I believe other readers will wonder the same thing.
2. Related – why not just try to decode the experiment phase using data from the neutral blocks? I believe this would work equivalently well, although there may be fewer trials. In general, I was surprised the authors never used EEG data from the neutral blocks – either for comparison to the biased blocks, ruling out the temporal confound, or testing how long the biased priority map representation is maintained.
3. I like the analysis decoding the previous target location. However, it seems like this analysis is always performed on the high-probability trials – why not also include the neutral trials? These trials should have no possible bias in previous target probability, which is present in the biased blocks (and addressed capably in Fig. 3B). Additionally, the authors should specify if this decoding is based on 8 or 4 alternatives (since now there are 8 possible target locations, rather than the 4 possible biased target locations).
4. I'm having trouble following Fig. 4A. The authors show a typical SDT plot (2 overlapping distributions with a set of criteria), and label them "Cell activation profile" in the figure (but "proposed tuning profile(s)" in the caption). The different distributions are, I presume, activity distributions across trials when a stimulus is the target vs distractor – suggesting that the cell can change its activity gain as a function of stimulus identity (at a relatively high level – target & distractor are defined as singletons, and the feature value for the target updates on each trial..). The threshold/criterion is explained to change as a function of learned target location probability, and is presumed to account for differences in behavioral performance and ping-evoked activity. I'm confused, though – what is the threshold/criterion doing here? Is this the spike threshold for a neuron? That wouldn't make much sense, since the activity distribution on the x axis is already scaled with target/distractor presence, which (to me) implies some degree of spiking

already...unless the x axis is meant to communicate the input to the neuron? Also, where is the distribution for stimuli that are non-distractors/non-targets? I presume it would be below both of these, but this isn't specified. Finally, I think the labels on the two thresholds (HP/LP) are backwards – wouldn't the HP threshold be lower (so a ping evokes greater activity) than a LP threshold? Now, after writing all these questions out, I think I can save the authors some time – I don't think this panel is necessary at all and could be removed. The rest of the figure is neat, and I think does a really clear job visualizing what the authors think is going on. I think it could actually work well as a motivation figure prior to showing results – but the authors should put it where they prefer of course!

Minor:

1. Is the 4-way decoding of biased target location primarily driven by the left/right locations, or do all 4 contribute? The authors could plot a confusion matrix for a window of the decoding timecourse to establish the spatial 'precision' of this effect.
2. End of 2nd paragraph on pg 5 (sentence starting "Furthermore, in line with...") – I believe the authors mean to communicate that performance is slower on trials when the distractor appeared in the biased high-probability target location – the sentence, as written, is confusing and could be interpreted to mean there is an uneven distribution of possible distractor locations
3. Related, is the location of the distractor at all biased during the biased blocks? It appears the authors fully randomized the distractor position with the only restriction being that the distractor is not the same stimulus as the target. But, if the target location is biased, would this result in a (slight) bias in the distractor location? If so, to what extent does any contribution from the learned distractor bias explain the results reported here?
4. Is there a difference in behavioral performance (RT/acc) as a function of ping presence? (and ping feature with respect to target feature on that trial: matching/non-matching?)
5. Why are different shapes used for the ping stimulus? The motivation behind this design decision is not mentioned in the manuscript. Reading between the lines, I imagine the authors were interested in whether the most-recently-selected target shape was represented differently from the other shape. Did the authors explore this possibility? If so, for completeness, the authors should report these results.
6. The preregistration describes ERP analyses on the N2PC component, but these do not appear to be reported in the manuscript. For completeness, the entire preregistration plan should be followed and reported (could go in Supp materials if necessary).
7. The authors should include a data availability statement in the manuscript (included in the reporting summary document)

Reviewer #2:

Remarks to the Author:

The authors show that the "ping" technique can be used to decode high probability targets during the ITI, and suggest that this provides evidence that the attentional priority map is mediated by latent neural mechanisms.

While the authors show a straightforward application of the "ping", I have some concerns about the analyses and the conclusions drawn

1. The authors acknowledge that there is a clear temporal confound due to autocorrelation in the signal, and run a "dummy" decoding analysis that seeks to control for this. However, I believe that this control analysis is not enough.

The comparison between the actual and the dummy decoding results is not fair, since fewer trials and "classes" are included in the latter. I propose the following instead: For the HP target decoding, use two separate classifiers, one using only the first three blocks, and one using the last three (so only three classes in each). Then take the average of the two. Next, use the full range of possible "dummy" decoders using separate classifiers for each (i.e. class 1 of dummy decoder 1 should include the 2nd to last biased loc. 1 block as well as the neutral block, and so on). In total you will have four dummy decoders. Take the average of the results. Furthermore, I propose to correct for this temporal confound outright by subtracting the average dummy decoder output from the HP decoder output.

Related to above: Aren't the results shown in Figure 3a and 3b also affected by the temporal

confound, since the location of the previous trial is more likely to be the HP target? If so, then this should be taken into account.

2. The authors show that the ITI seems "silent" since no HP decoding is present when no "ping" is presented. The employed analysis is at the very least incomplete, however. The authors take an arbitrary baseline in the no-ping condition during the ITI, and find that after this baseline (i.e. the voltage signal is set to 0), the signal does not produce a decodable signal within the next 600 ms. Given that the signal is likely very stable during the delay, this is hardly surprising, since the voltage trace was set to 0 and will stay there for a while. Indeed, even if there was a decodable signal in the stable activity during the ITI, this couldn't be picked up since that signal was removed by baselining. Nonetheless, baselining is important, due to drift. What I propose is to set the baseline prior to the array onset of the previous trial.

Additionally, and potentially more appropriate, is to use alpha power for decoding (commonly used for spatial WM decoding, e.g. Foster et al.), which does not require a baseline.

3. The authors used eye-tracking to control for systematic eye-movements. The eye-tracking signal is baselined before the onset of the ping, with the assumption that participants were looking at fixation. But exactly this should be tested. The data should not be baselined, instead it should be tested if gaze was systematically biased towards the HP target during the entire ITI in general, as well as just prior to impulse onset specifically. Systematic gaze shifts prior to ping would result in systematic offsets of the impulse stimulus, which could explain the decoding. The current analysis only tells us if there were systematic eye-movements shortly after impulse onset, not during or before. I furthermore suggest to not only take gaze position into account, but also systematic gaze shifts, as measured from the velocity of micro-saccades (e.g. Liu et al., Nature Communications, 2022)

Minor comments:

Typo in abstract, p.1, line 6, engender->engage (?)

p. 3, 2nd to last paragraph: It is unclear what the authors mean by "the latent attentional priority map was imaged in a neutral way". What does "neutral" mean here?

It would be nice to see if the ping had a behavioural effect.

The analysis related to figure 3b is unclear: The caption states that "decoders are trained only on..." Does this mean that the decoders were trained on a subset of trials, but tested on all of them? Or does this mean that only the subset of trials was included for both training AND testing?

Reviewer #3:

Remarks to the Author:

This manuscript is exceptional and I believe it deserves to be published in top journal with minimal revision. I have kept my eye on the pinging studies in working memory, but even though I'm fond of saying that most working memory research is really just attention research, I never thought to use the pinging approach to study attention directly. The particular application here is notably timely: several recent studies have failed to find neural signatures of selection history effects, and this new work squarely shows why. But more importantly, in my opinion, the basic idea of applying the pinging technique beyond working memory, and using it as a general tool for studying any conceivable class of potentially "silent" neural effects is exciting, and is likely to have a substantial impact on many areas of vision science and neuroscience. I have only a couple minor comments that I hope might help the authors to compose a final draft.

- AUC is a nice metric of decoding performance, but I do not believe it really conveys how well the decoder is doing. E.g. an AUC of .55 is not the same as 55% accuracy in a four-choice decoding problem (which I believe is the case here). I think being above chance (and the no-ping condition) is interesting enough in its own right, but it would be nice to have some idea of how well the decoder is doing: although decoding the prioritized location slightly above chance is important, the

strength of this signal is also relevant (e.g. 55 vs. 80% decoding accuracy). Or rather than accuracy, with some assumptions AUC can be converted to d-prime, which also provides a nice standard metric.

- On p. 5 it's noted that excluding 11 participants did not change the behavioral or decoding analyses, but then notes a significant effect. Presumably this refers to a significant decoding difference between ping and no-ping trials that remains for this subset of participants, but it isn't clear as written.

- EEG is not mentioned in the title or abstract, and only in passing toward the end of the introduction. It might be useful to note somewhere more prominent that this is an EEG study.

- I believe the authors that significant decoding of eye position is perhaps an artifact. However, from the current explanation it is really not clear how this is possible. It is suggested (p. 9) that above-chance eye tracker decoding "likely represents temporal noise in the eye tracking data". But my understanding of "noise" implies that it can not possibly support above-chance decoding. That is, if it is random noise any information in it should be orthogonal to the decoded features, and thus not support decoding. It's a relief that this decoding works for both ping and no-ping trials, and for "dummy" labels. But it's still weird, and any further explanation of what structured noise in the data might be causing it would be helpful.

- The authors do a nice job of showing that priming from the previous trial does not solely support decoding here. I have never used this task, however in similar RT tasks I have observed trial-to-trial priming as far back as three trials. For example, in a Stroop task RT on the current trial is pretty greatly affected not only by the congruent/incongruent condition of the previous trial, but also the one before that. I don't think this consideration warrants further analyses, which would probably be impossible as examining 2-back or 3-back effects requires a lot of data. And perhaps this has been performed and ruled out in previous behavioral work. However, it seems worth considering that the appearance of "selection history" effects or "statistical learning" in the present study might be more short-lived than the block level, and may reflect carryover from previous trials beyond the immediately preceding one. But even if that completely explained such effects, this would still be an interesting an important finding, probably just slightly different language and framing of the cause of the effect than a more global task-set effect that persists across a block.

We appreciated your comments, and were very pleased to read that you found our findings novel and exciting. We are happy to resubmit our revised manuscript incorporating your feedback, and we believe this new version greatly benefits from your input. To facilitate the review process, we have color-coded this response letter so that the **reviewers' comments are colored grey, our responses are colored black, and modifications to the manuscript are colored red**. In the revised manuscript all modifications are also colored red.

Quite a few novel analyses we're requested, resulting in a long response letter. We were very happy to run these analyses and in numerous occasions they led to valuable new additions to our manuscript. When, on a few occasions, we opted not to follow your advice, extensive justification was given as to why. Furthermore, while very interesting, in some cases we felt that the proposed additional analysis would distract from the main message of the manuscript, and so we have chosen not to include these new figures in the main text. Despite this, we are happy that Nature Communications uses a transparent review system, meaning that even though some of these figures may not be present in the main text, those really interested will have access to these analyses by way of these public reviews. We feel this is a welcome advantage of transparent reviews, acting as a sort of supplement to the supplements section.

We are sincerely grateful for your time in considering our article.

With regards,

Dock Duncan
Dirk van Moorselaar
Jan Theeuwes

Reviewer #1:

When we frequently perform a visual task with statistical regularities (e.g., looking for our keys on the way out the door; the keys are usually on the kitchen table, but sometimes your toddler moves them to the chair), behavior improves in its speed and accuracy based on the use of these regularities. In this manuscript, the authors test the possibility that spatial regularities learned during a visual search task are instantiated in a ‘silent’ adjustment of synaptic weights that support neural priority maps of visual space. They adopt the ‘pinging’ paradigm frequently used in visual WM experiments (e.g., Wolff et al, 2015) to measure the impact of a putative latent state of the neural priority map on visual processing of task-irrelevant visual stimuli. They find that, when a location is statistically more likely to contain a target stimulus, this location can be decoded based on evoked EEG activity patterns in response to a task-irrelevant ping stimulus. The authors conduct some important control analyses to rule out contributions from, e.g., eye movements. Overall, the authors conclude that learned probabilities of target locations on a visual search task are neurally instantiated via activity-silent mechanisms, consistent with a rapid reshaping of synaptic weights.

I really like this paper – the idea is simple but profound, and is one of those that made me think “how did I not think of that?!” The results are intriguing, and will certainly be of substantial interest to the readership of Nature Communications. I have several comments about the analyses presented by the authors (and the preregistered analyses), and other aspects of the manuscript. I expect these can be addressed with the data in-hand.

We greatly appreciate your words, especially that you find our results intriguing and we hope that this is a sentiment shared by many of the readership of Nature Communications. Many of the your comments have resulted in new content in the manuscript, and those that did not make it we are happy will still be available here in the public reviews as they pertain to interesting questions nonetheless.

1. I’m having a bit of trouble understanding the “dummy decoding” analysis illustrated in Fig. 1, and reported in Supp. Fig. 3A. The goal of this analysis is described as ruling out the possibility that subtle changes in ERPs over the course of the experiment session could explain the ping-evoked decoding observed independent of learned spatial regularities. The authors thus sample contiguous blocks of trials that span 2 blocks of one biased location, a neutral location block, and 1 block of another biased location. The authors describe in the caption of Fig. 1B that the authors use “fake category labels” – I’m interpreting this to mean that they arbitrarily assign each of the 3 dummy sets of blocks a number (1-3) and attempt to decode this value. The authors motivate the use of 4 adjoining blocks based on the importance of matching trial numbers, but the current scheme seems like it would still result in a biased sample of trials (more trials with 1 biased target location than any other). Could the authors instead sample 1.5 blocks before/after the neutral blocks? Perhaps the lack of significant decoding observed in Supp Fig. 3A suggests that this concern is moot, but I thought I’d raise it, as I believe other readers will wonder the same thing.

We thank the reviewer for this interesting suggestion. As displayed below, decoding following the regime suggested by the reviewer resulted in virtually identical results as our original dummy decoding regime (see figure below). We also considered this decoding scheme, but in the end, we decided our initial selection was slightly better for the reason that the neutral blocks are arguably not truly neutral. While behaviorally there was little to no evidence that participants continued to prioritize the previous high-probability location in the neutral blocks (and the decoding also shows no ability for our decoder to discriminate between our neutral blocks; see our response to comment 2), it should be noted that this comparison was based on relatively low trial counts. Therefore, it’s possible that lingering effects were obscured, especially since we know that (in the case of learned distractor suppression at least) learned attentional biases can continue to be present 100 trials after the removal of a regularity. Thus, to take these lingering attentional biases into account we choose to tip the balance of our trials slightly in favor of the second condition. This is now clarified on page 7.

Figure Caption: Dummy Decoding when centered around neutral blocks. Included neutral blocks, as well as 1.5 blocks before and after neutral blocks.

AMMENDED TEXT

(page 7) *Additionally, we chose to make this window overlap slightly more with the subsequent high-probability condition than the preceding one as there was a chance that participants would be in the act of “un-learning” the previous high-probability location for some time after the regularity was no longer present. We sought to proactively counteract such lingering biases by sliding the dummy window more in favor of the second high-probability condition.*

2. Related – why not **just try to decode the experiment phase using data from the neutral blocks?** I believe this would work equivalently well, although there may be fewer trials. In general, I was surprised the authors never used EEG data from the neutral blocks – either for comparison to the biased blocks, ruling out the temporal confound, or testing how long the biased priority map representation is maintained.

The reviewer is indeed correct in pointing out that we have done little with our neutral blocks. The main purpose of including these blocks was to reset the priority landscape such that when we introduced a new high probability location, learned attentional biases at preceding high probability locations would already be eliminated, or at least attenuated. As a result, these conditions contained relatively few trials (56 trials per the three neutral blocks before exclusions), which were potentially ‘confounded’ by lingering biases from the preceding block (see response above). This meant that if using data from neutral blocks only, we found any decoding, we might attribute it to a carryover effect, and if we found no decoding, then we might attribute it to a low trial count, thus making any resulting interpretation fallible. As a result, we chose to generally stay away from analyses focused on the intermediate “neutral” blocks. Having addressed this question by the reviewer, we also realized that the warning we give in the figure caption of supplementary figure 1 bringing attention to these low trial counts is perhaps not clear enough, and have added some words to make this warning clearer.

Nevertheless, we are happy to report these results here with the appropriate caveats. As can be seen, the decoding results are very similar to our dummy decoding results, but we believe that with the extremely limited trial number then it would have taken an exceptionally strong effect to produce any decoding in this condition anyway.

Figure Caption: Decoding of three neutral blocks.

AMMENDED TEXT

(Supplementary Figure 1 Caption) *Note that these averages were calculated on very small trial counts (as illustrated by the narrow bar widths) and so there remains the possibility that an effect exists in these blocks that our analyses were not sensitive enough to detect.*

3. I like the analysis decoding the previous target location. However, it seems like this analysis is always performed on the high-probability trials – **why not also include the neutral trials?** These trials should have no possible bias in previous target probability, which is present in the biased blocks (and addressed capably in Fig. 3B). Additionally, the authors should **specify if this decoding is based on 8 or 4 alternatives** (since now there are 8 possible target locations, rather than the 4 possible biased target locations).

Thank you for pointing this out. In review we realized we never stated explicitly that we included neutral trials in the previous trial target location decoding analysis (shown in figure 3A). We have adapted our transcript to reflect this. Furthermore, we now also clarify that this analysis was based on all 8 possible locations, which was also the case but not clearly stated.

AMMENDED TEXT

(page 7) *This analysis included all eight possible target locations and additionally included the neutral blocks.*

4. I'm having trouble following Fig. 4A. The authors show a typical SDT plot (2 overlapping distributions with a set of criteria), and label them “Cell activation profile” in the figure (but “proposed tuning profile(s)” in the caption). The different distributions are, I presume, activity distributions across trials when a stimulus is the target vs distractor – suggesting that

the cell can change its activity gain as a function of stimulus identity (at a relatively high level – target & distractor are defined as singletons, and the feature value for the target updates on each trial..). The threshold/criterion is explained to change as a function of learned target location probability, and is presumed to account for differences in behavioral performance and ping-evoked activity. I'm confused, though – what is the threshold/criterion doing here? Is this the spike threshold for a neuron? That wouldn't make much sense, since the activity distribution on the x axis is already scaled with target/distractor presence, which (to me) implies some degree of spiking already...unless the x axis is meant to communicate the input to the neuron? Also, where is the distribution for stimuli that are non-distractors/non-targets? I presume it would be below both of these, but this isn't specified. Finally, I think the labels on the two thresholds (HP/LP) are backwards – wouldn't the HP threshold be lower (so a ping evokes greater activity) than a LP threshold? Now, after writing all these questions out, I think I can save the authors some time – **I don't think this panel is necessary at all and could be removed.** The rest of the figure is neat, and I think does a really clear job visualizing what the authors think is going on. I think it could actually work well as a motivation figure prior to showing results – but the authors should put it where they prefer of course!

We appreciate this comment from the reviewer and we have taken a moment to think about what we really want to communicate with the figure and have come to the same conclusion that it is in fact best to exclude this panel. The reviewer was not the first person to express concern over its meaning, and we believe it distracts from what is actually a relatively simple message we mean to convey through the figure. Furthermore, perhaps an SDT graph is not the ideal figure for illustrating the cellular level changes we believe underly our observed effect. In any case, we have changed the figure as suggested and believe it does a better job of communicating our ideas in a simpler form.

NEW FIGURE 4

Minor:

5. Is the 4-way decoding of biased target location primarily driven by the left/right locations, or do all 4 contribute? The authors could plot a confusion matrix for a window of the decoding timecourse to establish the spatial 'precision' of this effect.

Thank you for pointing out this relevant point. The confusion matrix is shown below, taken for the time period from 300-400ms after ping onset, the window that consistently showed the highest decoding across all our various analyses. The numbers are the group average percentage difference from chance (25%) so a positive value is above chance and negative value below. As can be seen, while left and right both contributed to decoding, so did the top and bottom conditions.

We felt that this additional information was informative enough to warrant inclusion in the manuscript, and thus we have also included this confusion matrix as a panel in supplementary figure 5 in the main text. We thank the reviewer for this useful suggestion.

ADDED FIGURE

Predicted⇒ Actual⇓	Bottom	Left	Top	Right
Bottom	3	-1.4	-2	0.4
Left	-1.3	2.4	-0.2	-0.9
Top	-1.5	0	2.1	-0.7
Right	0.9	-1.4	-0.9	1.4

AMMENDED TEXT

(Figure 2 caption) *Post-hoc analysis additionally showed that all four locations contributed to this above chance decoding (see Supplementary Figure 5 for decoding without boosting, as per preregistration, as well as the confusion matrix).*

6. End of 2nd paragraph on pg 5 (sentence starting “Furthermore, in line with...”) – I believe the authors mean to communicate that performance is slower on trials when the distractor appeared in the biased high-probability target location – the sentence, as written, is confusing and could be interpreted to mean there is an uneven distribution of possible distractor locations

We appreciate the reviewers bringing this to our attention. We have rewritten it so as to ensure there is no confusion and it is clear that when distractors appear where targets are expected, they become more distracting.

AMMENDED TEXT

(Page 5) Additionally, in line with selective changes in attentional priority as a function of the introduced statistical regularities, trials in which a distractor happened to appear in the current high probability target location had especially slow response times, indicating that distractors interference was more pronounced when the distractor appeared in a location participants had been trained to expect targets ($t(23) = 3.442, p = 0.002, d_z = 0.7$; Figure 2C).

7. Related, is the location of the distractor at all biased during the biased blocks? It appears the authors fully randomized the distractor position with the only restriction being that the distractor is not the same stimulus as the target. But, if the target location is biased, would this result in a (slight) bias in the distractor location? If so, to what extent does any contribution from the learned distractor bias explain the results reported here?

We also appreciate the reviewer spotting that we never properly addressed this concern in the text – in fact in the experimental design we ensure that the distractor appears in all locations equally frequently to ensure that our effect is driven only by target locations. We have corrected this omission in the methodology section.

AMMENDED TEXT

(Methods, page 14) Distractors presentation was controlled such that a distractor appeared in each of the eight locations exactly the same number of times per block.

8. Is there a difference in behavioral performance (RT/acc) as a function of ping presence? (and ping feature with respect to target feature on that trial: matching/non-matching?)

This is again a very good question which we were also curious about – specifically we believed that the activation of the latent priority map by the ping may lead to an exaggerated speedup effect for high probability trials. What we found instead was that participants reliably responded faster on trials following a ping regardless of whether the target was at an HP or an LP location. We reasoned that these results were likely due to the ping acting as a sort of marker for temporal preparation – where participants knew that following a ping the search trial was to onset imminently (in the next 400-600ms). They could thus focus their attention in the moments immediately following the ping. On no-ping trials, on the other hand, no such warning sign was given and thus participants were less prepared at search display trial onset, leading to reliably slower reaction times. These results are shown below. Importantly, an ANOVA taking ping/no-ping and HP/LP on reaction times showed no interaction between the factors, meaning that the speedup was independent of the statistical learning effect (the ANOVA results are shown, with ping present/absent coded as ‘ping’ and target at high/low probability locations coded as ‘hp’).

Additionally, we checked whether this speedup was influenced by the target shape matching the preceding ping's shape, and found this was not the case ($t < 1$).

While this ping-evoked speedup effect is interesting, we don't feel it adds very much to our main findings, especially as it is equal between HP and LP target conditions in the following trial, and so have chosen not to include it in the main text.

Within Subjects Effects

Cases	Sum of Squares	df	Mean Square	F	p	η^2
ping	9509.920	1	9509.920	17.903	< .001	0.086
Residuals	12217.126	23	531.179			
hp	62163.658	1	62163.658	115.188	< .001	0.563
Residuals	12412.412	23	539.670			
ping *	214.112	1	214.112	0.355	0.557	0.002
hp						
Residuals	13859.903	23	602.604			

Note. Type III Sum of Squares

Figure Caption: Reaction Times in HP (left) and LP (right) conditions. Data is further split into trials in which a ping occurred immediately before search (orange) or the screen simply remained blank until the search trial onset (blue). The repeated-measure ANOVA is shown on the right panel showing the 2x2 design taking target location (HP/LP) and Ping presence (present/absent) as factors. As can be seen, significant main effects were found for both target location and ping presence, but there was no interaction.

9. Why are different shapes used for the ping stimulus? The motivation behind this design decision is not mentioned in the manuscript. Reading between the lines, I imagine the authors were interested in whether the most-recently-selected target shape was represented differently from the other shape. Did the authors explore this possibility? If so, for completeness, the authors should report these results.

The reason we initially designed the shapes as such was simply because other pinging studies had matched their ping stimulus to appear very similar to the working memory contents. We have now made this reasoning explicit when we discuss the experimental design in the methods section. It has been shown, however, that the ping does can also be dissimilar to the memory item in its features, as cross-modal decoding has also been shown using pings (where an audio memory is decoded using a visual ping – see Wolff et al., 2020). Regardless, to address the reviewers remark, we analyzed how the ping shape affected decoding and the post-hoc analyses shown below basically indicates that that there is no effect of matching versus mismatching shapes. Therefore, we decided not to discuss this in the main transcript.

Figure Caption: Shown are two attempts to decode shape information in our EEG data. In the figure to the left we show decoder accuracy when trained on the shape of the ping stimuli. As can be seen, our decoders were unable to consistently identify the shape of the ping when it was on the screen. On the right we show two attempts to decode the target shape in the previous trial. For the green line, we attempted to decode the preceding trials target shape using only trials in which the current ping matched that target shape. In contrast, for the orange line, we used only trials in which the target was a mismatch. In both conditions, the results never reached any level of significance, indicating that there was little/no meaningful shape information in our ongoing EEG signal.

AMMENDED TEXT

(Methods, page 14) *We chose to use pings that closely resembled the search targets in location and shape as this is typically done when this procedure is applied investigating working memory (Wolff et al., 2015, 2017; Wolff, Jochim, et al., 2020) (but see Rose et al., 2016; Wolff, Kandemir, et al., 2020 for exceptions).*

10. The preregistration describes ERP analyses on the N2PC component, but these do not appear to be reported in the manuscript. For completeness, the entire preregistration plan should be followed and reported (could go in Supp materials if necessary).

Thank you for pointing this out. We planned to analyze lateralized evoked activity to explore whether any ping-based decoding could be explained by covert attention towards the high probability location induced by the ping display. As visualized below, while there was a clear target elicited N2pc during visual search, no such lateralized activity was observed during processing of the ping display. This shows that our decoding cannot be attributed to participants accidentally attending to the ping stimuli at the high probability location. Nevertheless, the reviewer is entirely correct in saying that we should be consistent with our preregistration, and so we have included these figures in the supplements. This is now clarified on page 6, where we refer to the new figure in the supplementary material.

AMMENDED TEXT

(Page 6) *(see also Supplementary Figure 6 for an additional preregistered ERP analysis)*

(Methods, page 19) **ERP Analysis:** *An additional event related potential (ERP) analysis was conducted to investigate whether the ping displays produced a lateralized evoked component, such as the N2pc, which is known to indicate lateral attentional capture¹⁰⁴⁻¹⁰⁶. While the results did not produce anything of real interest, this analysis was included in the preregistration and thus is included as Supplementary Figures 6 for completeness. Two separate ERP analyses were conducted for ping and search displays. Data was 30 Hz low-pass filtered and baseline corrected using a 200ms window prior to ping or search onset. We first investigated whether an N2pc component could be found during search as a sanity check. Trials were thus restricted in our search analysis to those in which the target appeared on the left or right side of the horizontal midline. Furthermore, interference from the distractors was controlled by restricting trials of interest to those in which the distractor was present on the vertical meridian or trials in which no distractor was presented at all. For our analysis of the ping data, the primary factor of interest was block HP conditions, so analysis was restricted to trials in which the HP location was either the left or right locations in space, as there was no reason to expect top and bottom conditions to produce a perpendicular lateralized component. For each analysis, waveforms were computed for ipsilateral and contralateral scalp regions relative to the relevant horizontal factor using the PO7/PO8 electrodes of interest. Data was then 30 Hz low-pass filtered and baseline corrected using a 200ms window prior to ping or search onset.*

NEW SUPPLEMENTARY FIGURE 6

SUPPLEMENTARY FIGURE 6 – comparing N2pc evoked components on search trials versus ping trials. Search trials included only trials in which the target appeared on the leftmost or rightmost location in the array – with the evoked response flipped such that the ipsilateral and contralateral side in sensor space were congruent on every trial. Ping trials include only trials in which the HP location was on the horizontal midline (left or right) with the same sensor flipping approach used to ensure matched contra/ipsilateral arrangement. As shown, a clear N2pc was observed on the search trials, but none was observed on the ping trials. These results indicate that neither the ping evoked a strong attentional response, nor that the decoded results were the result of an evoked N2pc.

11. The authors should include a **data availability statement** in the manuscript (included in the reporting summary document)

A data availability statement has been added at the end of the methods section

ADDED TEXT

(Page 18) **Data Availability:** *All data will be made publicly available on OSF (<https://osf.io/v7yhc/>) upon publication.*

Reviewer #2:

The authors show that the “ping” technique can be used to decode high probability targets during the ITI, and suggest that this provides evidence that the attentional priority map is mediated by latent neural mechanisms.

While the authors show a straightforward application of the “ping”, I have some concerns about the analyses and the conclusions drawn

Thank you for your careful reading and appraisal of our work. We address your concerns in detail below; several of your observations have resulted in new figures and paragraphs in the manuscript which we feel greatly improve our text.

1. The authors acknowledge that there is a clear temporal confound due to autocorrelation in the signal, and run a “dummy” decoding analysis that seeks to control for this. However, I believe that this control analysis is not enough. **The comparison between the actual and the dummy decoding results is not fair, since fewer trials and “classes” are included in the latter.** I propose the following instead: For the HP target decoding, use two separate classifiers, one using only the first three **blocks**, and one using the last three (so only three **classes** in each). Then take the average of the two. Next, use the full range of possible “dummy” decoders using separate classifiers for each (i.e. class 1 of dummy decoder 1 should include the 2nd to last biased loc. 1 block as well as the neutral block, and so on). In total you will have four dummy decoders. Take the average of the results. Furthermore, **I propose to correct for this temporal confound outright by subtracting the average dummy decoder output from the HP decoder output.**

We thank the reviewer here for bringing up an important observation – that the number of classes could possibly influence decoder accuracy. In the revised MS we address this by adding a new panel to Supplementary Figure 3 and slightly changing panel A in the same figure with a new comparison.

The reviewer correctly points out that while the main analysis dissociated between four different high probability locations (i.e., four classes), the dummy decoding regiment only included three classes. Note however that in both analyses, each class contained observations from four different experimental blocks and hence the number of trials in these analyses was matched (see our response to reviewer 1 comment 1 for further discussion on these selected blocks). To better match the number of classes between analyses, for high probability location decoding we now also present the average of two iterations, in which in one iteration we **excluded** the first four blocks of the experiment (so the first HP location), and in a second iteration we excluded the last four blocks (the last HP location). Critically, this new analysis which replicated the main pattern of results was matched with our dummy decoding in trial count, class count, and, importantly, also temporal structure (We only ran this analysis excluding the first or last HP blocks as excluding either of the middle HP blocks would mean that the new dataset would be stretched over a much larger time window than our dummy decoding, an important consideration given we are investigating temporal effects). Unfortunately, we did not fully understand the reviewers second suggestion – splitting up our dummy decoders to get four classes– but we also believe this is no longer necessary given that the described approach already matches the main and the dummy decoding analysis, both in terms of number of observations and number of classes.

As can be seen, this new decoding approach yields the same pattern of results as our main analysis. For the comparison between this new decoding approach and the dummy decoding: we like the idea of subtracting the dummy decoding from our actual decoding, and we believe that this is essentially what we do when we analyze the difference between dummy and actual decoding using a cluster-based permutation test (the dotted lines at the bottom of the figure). We have amended our figure to now include both the comparison between dummy decoding and our main analysis (with four classes) and the new analysis suggested by the reviewer (with three classes). These new results are shown as purple dots at the bottom of the figure. As can be seen, the clusters are smaller but roughly overlap with those from the comparison to our main analysis. We believe that this additional figure has improved the validity of this analysis, and we thank the reviewer for the suggestion.

NEW SUPPLEMENTARY FIGURE 3

Supplementary Figure 3 – Dummy decoding. A) decoder performance when trained on EEG data and passed dummy labels (see figure 1B for illustration of dummy decoding). Dotted red bars indicate clusters in which decoding significantly differed from matched decoding shown in Figure 2D. Dotted purple bars indicate the same significant differences but in comparison to Supplementary Figure 3C. Note: y-axis is matched to results in Figure 2D for comparison. B) decoder performance when trained on eye tracking data and passed dummy labels. C) HP decoder performance when trained on three classes rather than four; in our decoding results shown in Figure 2D, these decoders were tasked with selecting between four classes – each class representing one of the four high-probability target locations. In contrast, our dummy decoders shown in Supplementary Figure 3A were only tasked with selecting between three classes – fake groups centered around our three neutral blocks. To account for these fundamental differences between these two classification routines, we ran our original decoding on meaningful labels in two iterations – once excluding the first four blocks (the first high-probability conditions), and a second time excluding the last four blocks (the last high-probability condition). These decoders were thus trained on only three classes each, same as the dummy decoding. We then averaged the results of these two iterations to get the results shown here. These results are thus a more valid comparison to our dummy decoding than those shown in Figure 2D.

2A. The authors show that the ITI seems “silent” since no HP decoding is present when no “ping” is presented. The employed analysis is at the very least incomplete, however. The authors take an arbitrary baseline in the no-ping condition during the ITI, and find that after this baseline (i.e. the voltage signal is set to 0), the signal does not produce a decodable signal within the next 600 ms. Given that the signal is likely very stable during the delay, this is hardly surprising, since the voltage trace was set to 0 and will stay there for a while. Indeed, even if there was a decodable signal in the stable activity during the ITI, this couldn’t be picked up since that signal was removed by baselining. Nonetheless, baselining is important, due to drift. What I propose is to set the baseline prior to the array onset of the previous trial.

This is a very important point. On the one hand, when one applies a baseline, either in close proximity or as suggested by the reviewer more distant from the hypothetical ping onset time (see below), one can never rule out with certainty that there was no stable ongoing activity that contained information about the learned spatial probabilities. On the other hand, especially in the current experimental design where individual classes (i.e., high probability target locations) were grouped in separate experimental phases, baselining is critical to prevent that classification solely reflects drifts in the signal. These temporal artefacts make it very difficult to interpret steady state ongoing decoding over longer time windows.

To address these issues, in a series of exploratory analyses we tried several novel analysis approaches that would allow us to analyze the data without baselining. The rationale for pursuing non-baselined decoding was that shifting the baseline further back in time, as suggested by the reviewer, would still result in the removal of a stable neural signature reflecting the current high probability location if present. Moreover, sustained ERP activity (but not alpha band activity; see our response to comment 2B) becomes less reliable as time progresses from the pre-stimulus baseline period because of slow drifts in the EEG offset (Luck, 2014) and given the randomization of ping and no ping trials and the jittered post-ping interval it is virtually impossible to select a clean interval in the preceding trial. Thus, shifting the baseline earlier in time unfortunately contaminates the signal of interest and cannot be used to examine whether a decodable signal was actually present but removed by our baselining procedure. We therefore exploratively examined whether we could split our training and testing data in such a way that baselining was no longer necessary. The reasoning here was that we may be able to select separate the training and testing blocks such that the use of temporal noise was disincentivized (i.e., because there no longer was a clear dissociated temporal structure; see figure below). However, as shown below, while with baselining all of our explorative train/test splits mimicked the main pattern of results, without baselining in all cases the results were consistent with what we would expect with significant temporal coloration from our un-baselined signal (and this coloration even occurs when training takes place on equal numbers of temporally quite distant trials).

Because we were unable to find a sound method by which to train our decoders on un-baselined data (see below our explanation why in the current dataset alpha based decoding also requires a baseline), we decided instead that an extra paragraph highlighting these caveats was the best solution. Furthermore, we felt it was appropriate to add several sentences giving context to the discussion of persistent activity in our task. Specifically, we wanted to highlight that much of the fascination with persistent versus silent models comes from the field of working memory (where pinging has exclusively been used in the past). The reason why this has been an issue is because “active” models are very common in working memory research, making silent models more novel. However, in the field of statistical learning, this is essentially reversed, with few people proposing that our ability to learn regularities is mediated by persistent activity. The reason for

this is that statistical learning is known to be very durable and long term, whereas working memory is neither. Importantly, intertrial effects could still reasonably be modeled as being essentially 'active' in origin. However, electrophysiological rat studies on intertrial effects have not found little evidence for this, and have even recently found strong evidence for synaptic mechanisms(Barbosa et al., 2020).

AMMENDED TEXT

(Page 11) As decoding is by itself uninformative about the underlying neural representation (Naselaris & Kay, 2015; van Moorselaar & Slagter, 2020), the nature of these activity-silent states that the ping succeeds in visualizing remains an open question. In the field of working memory, the debate over why pings reveal otherwise hidden memory content has focused on the question of whether the decoded memories are encoded in truly latent networks, mediated by neuroplasticity, and which are then reactivated by the ping(Rademaker & Serences, 2017; Stokes, 2015; Wolff et al., 2017, 2021), or alternatively whether these memories are simply mediated by ongoing neural activity below a certain detection threshold, in which case, the ping would simply serve to reduce signal variance such that these states can be visualized(Bae & Luck, 2019; Barbosa et al., 2021; Schneegans & Bays, 2017). While these questions are ultimately better answered via neurophysiology, in the current case of ping-evoked decoding of learned attentional preparation, the existing literature favors the former interpretation, as selection history, whether in a form of statistical learning or intertrial priming, has generally exhibited none of the characteristics of active neural processes (Adam & Serences, 2021; Barbosa et al., 2020; Moorselaar & Slagter, 2019; van Moorselaar et al., 2020). **In fact (and in sharp contrast with the field of working memory) there exists no major theory of the underlying mechanisms of statistical learning that is explicitly built on a concept of sustained, continuous neural activity. This is because, while active traces may in principle be a plausible neural mechanism for driving intertrial priming over relatively short timescales, such active mechanism seem unlikely to drive statistically learned spatial biases which are known to persist over very long time periods ranging from minutes(Britton & Anderson, 2020; Duncan & Theeuwes, 2020; Valsecchi & Turatto, 2021), to weeks(Chun & Jiang, 2003; Jiang et al., 2013; Turatto et al., 2018). As a result, models of statistical learning have generally resembled models of long-term memory (where synaptic models are favored for their durability over time) rather than working memory (where active firing models are more normal) (Batterink et al., 2019; Ferrante et al., 2018; Fiser & Lengyel, 2019; Frost et al., 2015; Schapiro et al., 2017). Despite this, it is important to realize that the current results should not be taken as an outright refutation of any model of history modulated attentional bias which might propose ongoing neural activity as the central neural mechanism underlying the effect. While our results are consistent with a model where there is no meaningful ongoing activity in the intertrial window, due to our use of baselining and filtering, we cannot definitively rule out that there is some activity still present. While further work is needed to clarify this point, what remains clear is that the pinging approach offers a novel tool for studying the underlying changes occurring in the brain which allow for history mediated behavior to arise.**

2B. Additionally, and potentially more appropriate, is to use **alpha power for decoding** (commonly used for spatial WM decoding, e.g. Foster et al.), which does not require a baseline.

This is a very interesting suggestion that we also considered. Consistent with previous studies, a time-frequency analysis showed no systematic lateralization when the high probability location was on the horizontal midline. However, there is the possibility that a multivariate approach was able to reveal spatially specific information within the alpha band. Indeed, alpha power decoding as applied in previous studies adopting a forward encoding approach (e.g., Foster et al., 2016, 2017; Foster & Awh, 2019) does not require a baseline. We also believe that this tuning approach holds large promise to investigate learned attentional biases as it cannot only reveal that there is spatially specific information in the signal, but also reveal the underlying representation that drives classification. It should be noted however that previous studies using this approach (including our own) used a **mixed**-design and hence are not confounded by systematic changes in alpha-power throughout the experiment. As visualized below, global alpha power gradually increased throughout the experiment and a classifier trained on non-baselined alpha power hence resulted in artificially high decoding throughout the entire interval. Similarly, the forward encoding model, despite using a set of spatially specific basis functions, also capitalizes on this monotonic increase in alpha power and hence resulted in artificially sharp tuning curves far exceeding those ever reported in the literature. This temporal coloring of non-baselined alpha power decoding was confirmed as that the same analysis yielded to our dummy labels also resulted in artificially high decoding. Unfortunately, the present experimental design thus does not allow for an analysis approach without baselining and hence we cannot exclude the possibility that the ongoing signal did actually contain information about the current high probability location, but this information was removed by baselining. As outlined above we now address this in a new paragraph in the General Discussion.

Figure Caption overall alpha levels across the four quarters of the experiment. Error bars are within-subject corrected.

2C. Related to above: **Aren't the results shown in Figure 3a and 3b also affected by the temporal confound**, since the location of the previous trial is more likely to be the HP target? If so, then this should be taken into account.

Figure 3a shows decoding of the target location in the preceding trial. Counter to the main analysis, this decoding was based on all 8 target locations and included also the neutral blocks (as clarified in reviewer 1, question 3). In other words, during cross-validation exemplars from each class were randomly sampled throughout the entire experiment, and hence any above chance decoding could not be explained by a temporal confound in the signal, nor the higher probability that a location repeated at high probability locations. In this decoding, four of the eight locations were locations that could be a high-probability location, and the reviewer is quite right in bringing to our attention that these four locations may have had a subset of trials drawn from relatively close proximity, and so a note has been added in the figure caption mentioning this. It is highly unlikely this would have resulted in positive encoding, however, as the majority of trials in high-probability blocks were not in fact high-probability trials (62.5% of trials had the target at one of the low-probability locations). So a decoder trained on these features would create a huge number of false alarms, driving down the AUC. By contrast, 3b shows the main analysis again but separately for trials where the target in the preceding display appeared on a high probability location and when it did not. In other words, this analysis is identical to the main analysis, and the interpretation of the dummy control analysis thus also applies here. This is now clarified on page 7.

AMMENDED TEXT

(Figure 3 caption) This analysis also included trials that could have been at the current high-probability location (32% of total trials).

(Page 7) Together, these analyses suggest that the observed decoding was indeed driven by learned latent attentional biases in response to the high probability location manipulation, and the following analyses treat the results as such.

3. The authors used eye-tracking to control for systematic eye-movements. The **eye-tracking signal is baselined** before the onset of the ping, with the assumption that participants were looking at fixation. But exactly this should be tested. The data should not be baselined, instead it should be tested if gaze was systematically biased towards the HP target during the entire ITI in general, as well as just prior to impulse onset specifically. Systematic gaze shifts prior to ping would result in systematic offsets of the impulse stimulus, which could explain the decoding. The current analysis only tells us if there were systematic eye-movements shortly after impulse onset, not during or before. I furthermore suggest to not only take gaze position into account, but also systematic gaze shifts, as measured from the velocity of micro-saccades (e.g. Liu et al., Nature Communications, 2022)

The reviewer brings up an important point that systematic shifts in fixation center can also influence decoding. We did in fact base our initial analysis on un-baselined eye tracking data. However, we were advised through conversations with several eye tracking experts that we should use caution in interpreting eye tracking data collected over long periods of time, with no systematic method of baselining. The reason for this is that eye tracking signals tend to drift over time, both due to slight changes in participant posture, but also just due to contributions by the eye tracker itself. In our study we did not baseline after every block, but only “as necessary,” meaning whenever it was visibly clear to the experimenter that the eye tracker was no longer at consistently at fixation. On the one hand this benefits our un-baselined analysis as were we to recalibrate frequently, then any systematic drift would be incorporated into our new fixation center – essentially accomplishing the same thing as post-hoc baselining. On the other hand, this means that we retained a great deal of signal noise in our raw channel recordings, meaning any analysis based on this raw data could easily be an analysis based on random noise (see also reviewer 3 question 4 for further thoughts on this).

Nevertheless, the reviewer brings up a good point that in order to analyze systematic drift, un-baselined eye tracking data must be used. To investigate this, we borrowed again from the analyses in van Ede et al., (2019) and analyzed our gaze data for towardness. This method is very similar to that used in Liu et al, (it is the same dataset and several of the same authors), except that rather than isolating the window around microsaccades, it is calculated over the whole raw eye tracker signal, making it also sensitive to the systematic drifts in the eye tracker signal. This means that towardness can both capture systematic patterns in microsaccadic movements, and also overall visual drift (as confirmed through conversations with the senior author).

The specific methods for calculating towardness are included below (as taken from van Ede et al., 2019). As can be seen in the figures below, the towardness analysis did not show any systematic shifts on the group level. On the level of individual differences, it can be seen that some variety in eye resting position existed between participants. We were also curious whether this score predicted each individual’s decoding performance (as would be expected if eye position was driving decoding), so we ran a correlation analysis between average gaze position, and decoding accuracy in the time window between 300 and 400ms after ping onset (which is generally where we see the highest decoding accuracy). As can be seen, gaze bias does not predict decoding results. However, it could alternatively be the absolute difference in gaze position predicts decoding accuracy, not just its systematic bias towards the HP location (i.e., decoding could increase not just as eyes move closer to the expected target location, but when their resting position was very different between conditions). This can be investigated using the absolute values of the towardness scores – which then become an axis-restricted degree of difference score instead. This correlation analysis was also carried out and showed that the level of difference between fixation position between conditions also did not predict decoding accuracy. We take these results to indicate that participants who showed the largest gaze drift between conditions in our eye tracking data did not also show the highest decoding in our EEG data. We have added this towardness analysis to Supplementary Figure 4 and referenced it in the main text.

Figure Caption: Top-Left) aggregate towardness scores in the intertrial period. See van Ede et. al., 2019 Figure 2C for a direct comparison in a dataset in which systematic microsaccades are found. Top-Right) individual towardness scores. Note the noise at the beginning of the trial for some participants may represent blink correction artefacts, as participants were encouraged to blink only immediately after search ended. Bottom-Left) correlation between average towardness score and decoding accuracy on ping trials in the 300-400ms window post ping. Bottom-right) same as bottom-left except using the absolute towardness score, thus representing the magnitude of fixation difference between conditions rather than systematic eye drift towards the HP location.

AMMENDED TEXT

(Page 9) To build on the results of the heatmap analysis, we next calculated a 'towardness score' (Van Ede et al., 2019) to quantify how eye position changed systematically across time in relation to the high-probability locations. This score quantifies systematic gaze shifts as a numeric score, and is sensitive to both microsaccades as well as overall shifts in resting gaze position in eye tracking data (Van Ede et al., 2019). The results of this analysis are shown in Supplementary Figure 4A; eye deviations did not systematically differ from zero, indicating no systematic gaze bias towards (or away) from the high probability locations. Furthermore, to assess whether individuals whose eyes drifted more had higher decoding scores, a correlation analysis was also done between each participants towardness score and their decoder performance over the 300-400ms window post ping (where decoding was highest). This correlation analysis is shown in Supplementary Figure 4B, and shows that resting eye position did not predict decoder performance.

(Methods, page 18) **Eye Towardness Calculation:** The eye towardness analysis shown in supplementary figure 4A is based on that used in van Ede et al. (2019). In this preregistered analysis, un-baselined eye tracking data was used. Data was first epoched in a -700ms to 600ms window centered around (hypothetical) ping onset. This window extended from the earliest fixation onset to the earliest next search trial onset, and therefore represents the majority of the intertrial window on every trial. Trials were then split into ping and no ping trials to be analyzed separately. Following the methods of van Ede et al. (2019) Experiment 2, we next separated our trials into those in which the high probability location was on one of the vertical positions (top or bottom), or one of the horizontal positions (left or right). Separately for each participant, for every individual timepoint we then averaged the x-axis channel values for the horizontal trials and the y-axis channel values for

the vertical trials (see van Ede et al.(2019) eye-tracking analysis for further information) to get the average x or y-axis behavior through the intertrial period. Next, we subtracted the right from the left values, or the top from the bottom values, and divided by two to get a difference score, with positive values representing the average eye positional bias towards the high probability location, and negative values representing bias away. Next, we further divided these values by the actual possible target locations and multiplied by 100 to get a percentage deviation towards the target locations. Finally, we averaged between our vertical and horizontal conditions to get a general towardness score for ping and no-ping trials.

4. Typo in abstract, p.1, line 6, engender->engage (?)

we have changed the word to “induce” so as to use a less esoteric term

AMMENDED TEXT

(abstract) Using a task known to induce statistical learning of target distributions, in the current study we demonstrate that this otherwise invisible, latent attentional priority map can be visualized during the intertrial period using a ‘pinging’ technique in conjunction with multivariate pattern analyses on EEG recordings

5. p. 3, 2nd to last paragraph: It is unclear what the authors mean by “the latent attentional priority map was imaged in a neutral way”. What does “neutral” mean here?

By ‘neutral’ we mean only to say that the ping was task irrelevant. In most decoding experiments, evoked responses to task-meaningful stimuli are compared – whether that be a meaningful retro cue, an unexpected stimulus, an item to be memorized or a cue necessitating a response. In the case of ping evoked decoding, the evoked response is behaviorally irrelevant, and thus we thought to call it ‘neutral.’ We have amended the text slightly to make this clearer.

AMMENDED TEXT

(Page 3) Such a finding would both inform the neural mechanisms underlying history influences on attentional selection as well as represent the first time the latent attentional priority map was imaged in a neutral way using a task-irrelevant ping.

6. It would be nice to see if the ping had a behavioural effect.

See our response to reviewer one, point 8.

7. The analysis related to figure 3b is unclear: The caption states that “decoders are trained only on...” Does this mean that the decoders were trained on a subset of trials, but tested on all of them? Or does this mean that only the subset of trials was included for both training AND testing?

The reviewer here points out that this phrasing is confusing and problematic. These decoders were trained AND tested on a (properly cross-validated) subsets of trials which exclusively did/did not follow an HP trial. However, the wording did indeed make it seem like they were only trained on these subsets and then tested on the larger dataset. We have changed this sentence to make it clear that the training and testing were both conducted over these subsets.

AMMENDED TEXT

(Figure 3 caption) B) Decoding results when decoders are trained and tested only on ping epochs which followed a trial in which the target was at a HP location (blue) or excluding all such HP trailing epochs (red).

Reviewer #3:

This manuscript is exceptional and I believe it deserves to be published in top journal with minimal revision. I have kept my eye on the pinging studies in working memory, but even though I'm fond of saying that most working memory research is really just attention research, I never thought to use the pinging approach to study attention directly. The particular application here is notably timely: several recent studies have failed to find neural signatures of selection history effects, and this new work squarely shows why. But more importantly, in my opinion, the basic idea of applying the pinging technique beyond working memory, and using it as a general tool for studying any conceivable class of potentially "silent" neural effects is exciting, and is likely to have a substantial impact on many areas of vision science and neuroscience. I have only a couple minor comments that I hope might help the authors to compose a final draft.

We really appreciate your opinion on our work, and agree broadly that working memory and attention are often deeply intertwined. We also appreciate your familiarity with the current state of the field, and hope that others find these results timely as well. We have changed the paper's title in response to one of your comments, and we believe this new title will benefit the paper by priming readers for EEG data. Your comments have also given us an opportunity to speculate somewhat on our results, and we are very happy that these conversations will accompany our main script through these public reviews.

1. AUC is a nice metric of decoding performance, but I do not believe it really conveys how well the decoder is doing. E.g. an AUC of .55 is not the same as 55% accuracy in a four-choice decoding problem (which I believe is the case here). I think being above chance (and the no-ping condition) is interesting enough in its own right, but it **would be nice to have some idea of how well the decoder is doing**: although decoding the prioritized location slightly above chance is important, the strength of this signal is also relevant (e.g. 55 vs. 80% decoding accuracy). Or rather than accuracy, with some assumptions AUC can be converted to d-prime, which also provides a nice standard metric.

Perhaps the confusion matrix shown in the response to reviewer 1's fifth question is informative here (also added as a new panel in supplementary figure 5). As can be seen there, the probability of selecting the correct condition never really gets above a few percentage points over chance; so if the reviewer's question was whether we would be willing to bet our house on our decoders selecting the right HP location, our answer would be a resounding 'no.'

Despite this, this slight effect is highly consistent. This is perhaps comparable to how some attention effects are only a couple dozen milliseconds in length, it is ultimately not the absolute size of the effect that is of interest, but what its significance says about our underlying cognition. Furthermore, the current study is essentially a proof of concept, and we believe that future studies will improve upon our design, leading to better decoding results in the future.

2. On p. 5 it's noted that excluding 11 participants did not change the behavioral or decoding analyses, but then notes a significant effect. Presumably this refers to a significant decoding difference between ping and no-ping trials that remains for this subset of participants, **but it isn't clear as written**.

Upon reviewing this paragraph, we agree with the reviewer that this sentence could be better phrased to make its interpretation easier. We have amended it with the text below.

AMMENDED TEXT

(Page 5) When excluding these 11 participants, our remaining dataset continued to show strong target enhancement at the high probability location ($t(12) = 6.751, p < 0.001, d_z = 1.9$) suggesting that our observed results were not necessarily driven by a subset of aware participants (see Supplementary Figure 2 for the decoding results from the next section excluding these same 11).

3. EEG is not mentioned in the title or abstract, and only in passing toward the end of the introduction. **It might be useful to note somewhere more prominent that this is an EEG study.**

This is an interesting point; we realize that indeed it is important for readers to know this is an EEG experiment from the very beginning. As such we have chosen to change the paper title to make this expectation explicit; adding "using encephalography" to the end. We have also slightly changed the abstract to make this clearer.

NEW TITLE:

Pinging the brain to reveal the hidden attentional priority map using encephalography

AMMENDED TEXT

(Abstract) Using a task known to induce statistical learning of target distributions, in an EEG study we demonstrate that this otherwise invisible, latent attentional priority map can be visualized during the intertrial period using a 'pinging' technique in conjunction with multivariate pattern analyses

4. I believe the authors that significant decoding of eye position is perhaps an artifact. However, from the current explanation it is really not clear how this is possible. It is suggested (p. 9) that above-chance eye tracker decoding "likely represents temporal noise in the eye tracking data". But my understanding of "noise" implies that it can not possibly support above-chance decoding. That is, if it is random noise any information in it should be orthogonal to the decoded features, and thus not support decoding. It's a relief that this decoding works for both ping and no-ping trials, and for "dummy" labels. But it's still weird, and **any further explanation of what structured noise in the data might be causing it would be helpful.**

We share both the reviewer's confusion and relief with the observed pattern of results in regards to our eye tracking decoding. We have some suspicions on what is causing this distinct noise in our eye tracker signals, but these suspicions are entirely speculative at this point. The first thing to note is that eye trackers reliably add noise to the recorded signal – Niehorster et al. (2021) recently demonstrated this fact nicely in a study using artificial eyeballs. This study showed that even though the eye was most certainly not moving, some jitter in the signal was consistently recorded. It is our suspicion that the signal jitter added by the eye tracker might be influenced by subtle changes in the participants facial angle relative to the eye tracker. Our participants were allowed to remove their heads from the chinrest between blocks. The chin rest and fixation dot ensured that when they continued the experiment, their perceptual experience would be unchanged. The actual resting position of their face, however, could subtly change after these breaks. One possible effect of this is a slight change in the eyes relative angle to the eye tracker camera. We speculate that such subtle angular changes may then have had an effect on the nature of the noise the eye tracker was then adding to the signal. These changes in the jittering noise from the eye tracker may then have been picked up by our LDA decoders, leading our eyetracker-data-trained decoders to tell us essentially what angle the participants eye had relative to the eye tracker at that moment – a feature which had some temporally predictive value.

We further agree with the reviewer that this spurious decoding seems unlikely to be influencing our main EEG decoding. As a result, any further steps to investigate this effect would be superfluous to our main investigation, and probably of little interest to all but a very small number of people. As a result, we are comfortable leaving our suspicions at the level of speculation.

5. The authors do a nice job of showing that priming from the previous trial does not solely support decoding here. I have never used this task, however in similar RT tasks I have observed trial-to-trial priming as far back as three trials. For example, in a Stroop task RT on the current trial is pretty greatly affected not only by the congruent/incongruent condition of the previous trial, but also the one before that. I don't think this consideration warrants further analyses, which would probably be impossible as examining 2-back or 3-back effects requires a lot of data. And perhaps this has been performed and ruled out in previous behavioral work. However, **it seems worth considering that the appearance of "selection history" effects or "statistical learning" in the present study might be more short-lived than the block level, and may reflect carryover from previous trials beyond the immediately preceding one.** But even if that completely explained such effects, this would still be an interesting an important finding, probably just slightly different language and framing of the cause of the effect than a more global task-set effect that persists across a block.

The reviewer brings up an interesting point. We think important to firstly note that our effect can be observed even when trials have been set up such that no target location repetitions ever happen – meaning that statistical learning effects can still be observed even when repetition intertrial effects are greatly reduced in their influence (Goschy et al., 2014). Secondly, there are many other 'statistical learning' paradigms in which the learned regularity cannot be attributed to trial repetitions – such as in context cuing paradigms or intertrial target cuing, the second of these in which the intertrial effect is what is statistically learned (Chun & Jiang, 2003; Li & Theeuwes, 2020; Wang et al., 2021). A similar thing can also be said of triplet learning paradigms – a very different sort of statistical learning paradigm, but one where again the learned effect can be reduced to a sort of intertrial priming, just not one as simple as a repetition effect (Turk-Browne et al., 2005). Thus, this question also gets to the heart of whether the effect we observe here and call 'statistical learning' is really the same underlying mechanism as these other paradigms that also use this label.

We believe that we share the reviewer's pragmatic approach in that we don't see a real issue in saying that statistical learning and intertrial priming may really be the same thing, but expressed over different intervals and levels of complexity. A few repetitions would then have a short-lived boosting effect which quickly fades away, while a long period of performing a task with a statistical imbalance could lead to more permanent network changes in the brain, and changes which would persist across long time periods. The challenge then is making the jump between the relatively simple principles underlying the learning in our simple paradigm, to the extremely complex behavior necessary for the complex statistical relationships that we internalize in the real world (e.g., language syntax). This is clearly a topic for future work to come, and we believe we will have a working model for in the coming years. Furthermore, we believe that in the very near future we will be able to say quite a lot about how the specific neural processes whereby the brain internalizes past experiences to guide future action, and how these simple cognitive processes can aggregate to create extremely complex behavior.

References

- Adam, K. C., & Serences, J. T. (2021). History modulates early sensory processing of salient distractors. *Journal of Neuroscience*, *41*(38), 8007–8022.
- Bae, G.-Y., & Luck, S. J. (2019). Reactivation of previous experiences in a working memory task. *Psychological Science*, *30*(4), 587–595.
- Barbosa, J., Lozano-Soldevilla, D., & Compte, A. (2021). Pinging the brain with visual impulses reveals electrically active, not activity-silent, working memories. *PLoS Biology*, *19*(10), e3001436.
- Barbosa, J., Stein, H., Martinez, R. L., Galan-Gadea, A., Li, S., Dalmau, J., Adam, K., Valls-Solé, J., Constantinidis, C., & Compte, A. (2020). Interplay between persistent activity and activity-silent dynamics in the prefrontal cortex underlies serial biases in working memory. *Nature Neuroscience*, *23*(8), 1016–1024.
- Batterink, L. J., Paller, K. A., & Reber, P. J. (2019). Understanding the neural bases of implicit and statistical learning. *Topics in Cognitive Science*, *11*(3), 482–503.
- Britton, M. K., & Anderson, B. A. (2020). Specificity and persistence of statistical learning in distractor suppression. *Journal of Experimental Psychology: Human Perception and Performance*, *46*(3), 324.
- Chun, M. M., & Jiang, Y. (2003). Implicit, long-term spatial contextual memory. *Journal of Experimental Psychology: Learning, Memory, and Cognition*, *29*(2), 224.
- Duncan, D., & Theeuwes, J. (2020). Statistical learning in the absence of explicit top-down attention. *Cortex*, *131*, 54–65.
- Ferrante, O., Patacca, A., Di Caro, V., Della Libera, C., Santandrea, E., & Chelazzi, L. (2018). Altering spatial priority maps via statistical learning of target selection and distractor filtering. *Cortex*, *102*, 67–95.
- Fiser, J., & Lengyel, G. (2019). A common probabilistic framework for perceptual and statistical learning. *Current Opinion in Neurobiology*, *58*, 218–228.
- Foster, J. J., & Awh, E. (2019). The role of alpha oscillations in spatial attention: Limited evidence for a suppression account. *Current Opinion in Psychology*, *29*, 34–40.
- Foster, J. J., Sutterer, D. W., Serences, J. T., Vogel, E. K., & Awh, E. (2016). The topography of alpha-band activity tracks the content of spatial working memory. *Journal of Neurophysiology*, *115*(1), 168–177.
- Foster, J. J., Sutterer, D. W., Serences, J. T., Vogel, E. K., & Awh, E. (2017). Alpha-band oscillations enable spatially and temporally resolved tracking of covert spatial attention. *Psychological Science*, *28*(7), 929–941.
- Frost, R., Armstrong, B. C., Siegelman, N., & Christiansen, M. H. (2015). Domain generality versus modality specificity: The paradox of statistical learning. *Trends in Cognitive Sciences*, *19*(3), 117–125.
- Goschy, H., Bakos, S., Müller, H. J., & Zehetleitner, M. (2014). Probability cueing of distractor locations: Both intertrial facilitation and statistical learning mediate interference reduction. *Frontiers in Psychology*, *5*, 1195.
- Jiang, Y. V., Swallow, K. M., Rosenbaum, G. M., & Herzig, C. (2013). Rapid acquisition but slow extinction of an attentional bias in space. *Journal of Experimental Psychology: Human Perception and Performance*, *39*(1), 87.
- Li, A.-S., & Theeuwes, J. (2020). Statistical regularities across trials bias attentional selection. *Journal of Experimental Psychology: Human Perception and Performance*, *46*(8), 860.
- Moorselaar, D. van, & Slagter, H. A. (2019). Learning What Is Irrelevant or Relevant: Expectations Facilitate Distractor Inhibition and Target Facilitation through Distinct Neural Mechanisms. *Journal of Neuroscience*, *39*(35), 6953–6967.
<https://doi.org/10.1523/JNEUROSCI.0593-19.2019>
- Naselaris, T., & Kay, K. N. (2015). Resolving ambiguities of MVPA using explicit models of representation. *Trends in Cognitive Sciences*, *19*(10), 551–554.
- Niehorster, D. C., Zemblys, R., & Holmqvist, K. (2021). Is apparent fixational drift in eye-tracking data due to filters or eyeball rotation? *Behavior Research Methods*, *53*(1), 311–324.
- Rademaker, R. L., & Serences, J. T. (2017). Pinging the brain to reveal hidden memories. *Nature Neuroscience*, *20*(6), 767–769.
- Rose, N. S., LaRoque, J. J., Riggall, A. C., Gosseries, O., Starrett, M. J., Meyering, E. E., & Postle, B. R. (2016). Reactivation of latent working memories with transcranial magnetic stimulation. *Science*, *354*(6316), 1136–1139.
- Schapiro, A. C., Turk-Browne, N. B., Botvinick, M. M., & Norman, K. A. (2017). Complementary learning systems within the hippocampus: A neural network modelling approach to reconciling episodic memory with statistical learning. *Philosophical Transactions of the Royal Society B: Biological Sciences*, *372*(1711), 20160049.

- Schneegans, S., & Bays, P. M. (2017). Restoration of fMRI decodability does not imply latent working memory states. *Journal of Cognitive Neuroscience*, 29(12), 1977–1994.
- Stokes, M. G. (2015). 'Activity-silent' working memory in prefrontal cortex: A dynamic coding framework. *Trends in Cognitive Sciences*, 19(7), 394–405.
- Turatto, M., Bonetti, F., & Pascucci, D. (2018). Filtering visual onsets via habituation: A context-specific long-term memory of irrelevant stimuli. *Psychonomic Bulletin & Review*, 25(3), 1028–1034. <https://doi.org/10.3758/s13423-017-1320-x>
- Turk-Browne, N. B., Jungé, J. A., & Scholl, B. J. (2005). The automaticity of visual statistical learning. *Journal of Experimental Psychology: General*, 134(4), 552.
- Valsecchi, M., & Turatto, M. (2021). Distractor filtering is affected by local and global distractor probability, emerges very rapidly but is resistant to extinction. *Attention, Perception, & Psychophysics*, 83(6), 2458–2472.
- Van Ede, F., Chekroud, S. R., & Nobre, A. C. (2019). Human gaze tracks attentional focusing in memorized visual space. *Nature Human Behaviour*, 3(5), 462–470.
- van Moorselaar, D., Lampers, E., Cordesius, E., & Slagter, H. A. (2020). Neural mechanisms underlying expectation-dependent inhibition of distracting information. *Elife*, 9, e61048.
- van Moorselaar, D., & Slagter, H. A. (2020). Inhibition in selective attention. *Annals of the New York Academy of Sciences*, 1464(1), 204–221.
- Wang, L., Wang, B., & Theeuwes, J. (2021). Across-trial spatial suppression in visual search. *Attention, Perception, & Psychophysics*, 83(7), 2744–2752.
- Wolff, M. J., Akyürek, E., & Stokes, M. G. (2021). *What is the functional role of delay-related alpha oscillations during working memory?*
- Wolff, M. J., Ding, J., Myers, N. E., & Stokes, M. G. (2015). Revealing hidden states in visual working memory using electroencephalography. *Frontiers in Systems Neuroscience*, 9, 123.
- Wolff, M. J., Jochim, J., Akyürek, E. G., Buschman, T. J., & Stokes, M. G. (2020). Drifting codes within a stable coding scheme for working memory. *PLoS Biology*, 18(3), e3000625.
- Wolff, M. J., Jochim, J., Akyürek, E. G., & Stokes, M. G. (2017). Dynamic hidden states underlying working-memory-guided behavior. *Nature Neuroscience*, 20(6), 864–871.
- Wolff, M. J., Kandemir, G., Stokes, M. G., & Akyürek, E. G. (2020). Unimodal and bimodal access to sensory working memories by auditory and visual impulses. *Journal of Neuroscience*, 40(3), 671–681.

Reviewers' Comments:

Reviewer #1:

Remarks to the Author:

The authors have addressed all my comments, and I look forward to learning more about this exciting line of work in the future!

Reviewer #3:

Remarks to the Author:

In this revision Duncan et al. have done a nice job of addressing my comments on the previous submission, as well as those from the other reviewers. My concerns were largely minor and have mostly been addressed. Reviewer 2 had more substantive concerns, and I thought several points regarding potential side effects of the baseline procedures were thoughtful. But the authors have addressed these concerns by conducting several new analyses, and the conclusions still appear sound. The baseline procedures and some other aspects of the work might warrant further exploration, such as the observed above-chance decoding from the eye-tracking data. But I think the authors have done their due diligence and then some in this manuscript to provide convincing support for main conclusions, and I again think the work will have a substantial impact on the field.

My only remaining suggestion is to report the average decoder accuracy. In their reply the authors note that the new supplementary Figure 5 does so to some degree, but it's still pretty obscure. This supplemental figure shows a confusion matrix, but in units of deviation from chance. I calculated the average of the diagonal elements as 2.225, which I believe means that the average classification accuracy is $25 + 2.225 = 27.225\%$. I may or may not have done that right, but I don't think the reader should have to do their own calculations to figure out a pretty fundamental aspect of the data. I agree with the authors that they shouldn't bet their house on this level of classification performance, and agree that the fact that performance is above chance is nonetheless noteworthy. However, I do not think that the statistical significance of classification performance can be fully interpreted without some knowledge of the level of performance that led to the significant deviation from chance. As an analogy, a new drug might significantly extend the lifespan of cancer patients with a tiny p-value relative to a placebo. Although such a result might be important on its own, it's impossible to judge the impact of it without knowing by how many years, days, or minutes by which the drug improved lifespan. So while I agree that the significant classification result is important, I'm not sure it's a good idea to bet ones house on a significant p-value (which is what is being asked of the reader) but not on the size of the effect that caused it. As such, and as in my previous review, I think it's worth the ink to explicitly report average classification accuracy so that readers have some sense of how good the decoding is here. One option might be to report average accuracy initially, but then explain why this value might be difficult to interpret given its non-linearity relative to different chance baselines (e.g. 27% above a 25% chance baseline is not the same as 52% above a 50% chance baseline), and perhaps by noting that mean accuracy does not take some aspects of performance into account (like bias), whereas AUC addresses such issues. I think this would help motivate the use of AUC over raw accuracy as the dependent variable, while not making the relatively low classification performance detract from the importance of the consistently above-chance classification performance that is observed.

Reviewer #4:

Remarks to the Author:

Review of Duncan, van Moorselaar, & Theeuwes "Pinging the brain to reveal the hidden attentional priority map using encephalography"

Comments

The authors present a novel application of the impulse perturbation (pinging) method, to investigate 'hidden' attentional priority maps that are acquired over the course of blocks of trials

through statistical learning. Results show that the impulse response reliably reveals the learned expectation of the most probable target location. This is an important finding.

In this revised version of the manuscript (the original of which I did not see), the authors address a range of comments made by previous reviewers. Upon reading the manuscript, I was most concerned with the potential confounds of eye movements and temporal correlations, and with the discussion of the purported impulse effect on the activity-silent mechanisms that might underlie statistical learning.

With regard to eye movements, I think the towardness analyses presented by the authors in response to a previous reviewer comment are particularly convincing. Therefore I am not worried that these might have confounded the decoding. The temporal correlation problem is a bit more tricky, and it is probably impossible to get rid of it completely in a statistical learning paradigm such as this one. However, I do believe that by doing both the 3- and 4-class dummy decoding, again prompted by a previous reviewer comment, it is at least much less likely that decoding is driven by temporal correlations within blocks. The authors also discuss this appropriately in the text.

I do have two quibbles with regard to the discussion of activity-silent mechanisms. First, I think that the reason for why the authors are not analyzing alpha power to look for ongoing activity should be included in the manuscript, as this analysis would be expected by most readers. Second, I think the framing of the impulse effect in terms of "misfiring", as it is referred to in several places, is putting a subjective label on it. The impulse effect is certainly incidental, but given that it is known not to influence behavior, it seems too much to suggest that something is going wrong when it is delivered. Note that for TMS-based perturbation, "misfiring" might be more appropriate, but then again the TMS effect on WM memoranda is actually positive (i.e., results in better recall).

RESPONSE TO REVIEWERS

We would like to thank the reviewers for their positive appraisal of our work and their insightful commentary which has surely improved this final manuscript. Please find below our final responses to the reviewers comments. Reviewer remarks are shaded grey, our response is colored black and our in-text changes are colored red. Changes made in our final manuscript are also coded in red.

Thank you for your time and best regards,
Dock, Dirk & Jan

REVIEWERS' COMMENTS

Reviewer #1 (Remarks to the Author):

The authors have addressed all my comments, and I look forward to learning more about this exciting line of work in the future!

We thank the reviewer for their thoughtful and constructive comments.

Reviewer #3 (Remarks to the Author):

In this revision Duncan et al. have done a nice job of addressing my comments on the previous submission, as well as those from the other reviewers. My concerns were largely minor and have mostly been addressed. Reviewer 2 had more substantive concerns, and I thought several points regarding potential side effects of the baseline procedures were thoughtful. But the authors have addressed these concerns by conducting several new analyses, and the conclusions still appear sound. The baseline procedures and some other aspects of the work might warrant further exploration, such as the observed above-chance decoding from the eye-tracking data. But I think the authors have done their due diligence and then some in this manuscript to provide convincing support for main conclusions, and I again think the work will have a substantial impact on the field.

My only remaining suggestion is to report the average decoder accuracy. In their reply the authors note that the new supplementary Figure 5 does so to some degree, but it's still pretty obscure. This supplemental figure shows a confusion matrix, but in units of deviation from chance. I calculated the average of the diagonal elements as 2.225, which I believe means that the average classification accuracy is $25 + 2.225 = 27.225\%$. I may or may not have done that right, but I don't think the reader should have to do their own calculations to figure out a pretty fundamental aspect of the data. I agree with the authors that they shouldn't bet their house on this level of classification performance, and agree that the fact that performance is above chance is nonetheless noteworthy. However, I do not think that the statistical significance of classification performance can be fully interpreted without some knowledge of the level of performance that led to the significant deviation from chance. As an analogy, a new drug might significantly extend the lifespan of cancer patients with a tiny p-value relative to a placebo. Although such a result might be important on its own, it's impossible to judge the impact of it without knowing by how many years, days, or minutes by which the drug improved lifespan. So while I agree that the significant classification result is important, I'm not sure it's a good idea to bet ones house on a significant p-value (which is what is being asked of the reader) but not on the size of the effect that caused it. As such, and as in my previous review, I think it's worth the ink to explicitly report average classification accuracy so that readers have some sense of how good the decoding is here. One option might be to report average accuracy initially, but then explain why this value might be difficult to interpret given its non-linearity relative to different chance baselines (e.g. 27% above a 25% chance baseline is not the same as 52% above a 50% chance baseline), and perhaps by noting that mean accuracy does not take some aspects of performance into account (like bias), whereas AUC addresses such issues. I think this would help motivate the use of AUC over raw accuracy as the dependent variable, while not making the relatively low classification performance detract from the importance of the consistently above-chance classification performance that is observed.

We have made the requested changes to the format of supplementary figure 5 along with adding a few words in the figure caption on how to interpret these results. We would like to thank the reviewer for their positive and constructive feedback.

SUPPLEMENTARY FIGURE 5 – Supporting decoding figures. A) In order to increase the signal-to-noise ratio, techniques were borrowed from Grootswagers et al. (2017). These techniques were originally not planned for in the preregistration. The originally planned decoding without these boosting additions, in accord with the preregistered decoding pipeline, are shown in this figure. Note that the general pattern of results does not differ greatly in the timecourse of the overall pattern of decoding from that shown in Figure 2. Shaded areas represent standard error of participant means. B) Shown is the confusion matrix for the boosted decoding shown in figure 2 taken for the period of highest decoding (300-400ms post ping onset). Note that chance accuracy would be 25%. These numbers represent raw decoder accuracy, and thus should be interpreted with caution as they do not linearly translate to different chance baselines (the combined measure of AUC does a better job of representing standardized decoder performance as it combines multiple measurement thresholds).

Reviewer #4 (Remarks to the Author):

Review of Duncan, van Moorselaar, & Theeuwes “Pinging the brain to reveal the hidden attentional priority map using encephalography”

Comments

The authors present a novel application of the impulse perturbation (pinging) method, to investigate ‘hidden’ attentional priority maps that are acquired over the course of blocks of trials through statistical learning. Results show that the impulse response reliably reveals the learned expectation of the most probable target location. This is an important finding.

In this revised version of the manuscript (the original of which I did not see), the authors address a range of comments made by previous reviewers. Upon reading the manuscript, I was most concerned with the potential confounds of eye movements and temporal correlations, and with the discussion of the purported impulse effect on the activity-silent mechanisms that might underlie statistical learning.

With regard to eye movements, I think the towardness analyses presented by the authors in response to a previous reviewer comment are particularly convincing. Therefore I am not worried that these might have confounded the decoding. The temporal correlation problem is a bit more tricky, and it is probably impossible to get rid of it completely in a statistical learning paradigm such as this one. However, I do believe that by doing both the 3- and 4-class dummy decoding, again prompted by a previous reviewer comment, it is at least much less likely that decoding is driven by temporal correlations within blocks. The authors also discuss this appropriately in the text.

I do have two quibbles with regard to the discussion of activity-silent mechanisms. First, I think that the reason for why the authors are not analyzing alpha power to look for ongoing activity should be included in the manuscript, as this analysis would be expected by most readers. Second, I think the framing of the impulse effect in terms of “misfiring”, as it is referred to in several places, is putting a subjective label on it. The impulse effect is certainly incidental, but given that it is known not to influence behavior, it seems too much to suggest that something is going wrong when it is delivered. Note that for TMS-based perturbation, “misfiring” might be more appropriate, but then again the TMS effect on WM memoranda is actually positive (i.e., results in better recall).

We appreciate that the reviewer was able to evaluate our response to a previous reviewer’s comments; this is perhaps more difficult a task than providing your own feedback and we appreciate the care and speed in which they were able to do this. We have added a new reference to a new paper in which we revisit the current dataset and which includes a full analysis of proactive alpha (it presents a null finding which we felt only distracted from the current results but are interesting in regards to the new analysis’ research question). We have also amended the main text as

requested to replace the term ‘misfiring’ with ‘activating’ which we believe does a more neutral job of describing the underlying neural mechanism we hypothesize to be driving our observed effect.

Page 5

... (see also Supplementary Figure 6 for an additional preregistered ERP analysis. For a further analysis of pre-stimulus alpha using this dataset, see ⁵⁰).

Page 10

If latent learned attentional priority is mediated by neural structures entering a ‘primed’ neural state via processes of neural plasticity, then the passing of irrelevant but high-contrast pings through the visual cortex may incidentally activate these primed neurons at a high rate. This rate of activation, then, would be the weak signal to which the decoders are sensitive (Figure 4A).